# R-loop editing by DNA cytosine deaminase APOBEC3B modulates the activity of oestrogen receptor enhancers

Chi Zhang [1,2,3], Yu-jing Lu [4,7], Bingjie Chen [5,7], Zhiyan Bai[6], Qiaoxi Zeng[2,3], Alexia Hervieu [1], Marco P. Licciardello[1], Konstantinos Mitsopoulos[1], Bissan Al-Lazikani[1], Marcello Tortorici[1], Olivia W. Rossanese [1] ✉, Paul Workman [1] ✉ & Paul A. Clarke [1] ✉

Oestrogen receptor (ER) activation leads to the formation of DNA double strand breaks (DSB), promoting genomic instability and tumour heterogeneity. The single-stranded DNA cytosine deaminase APOBEC3B (A3B) serves as a co-activator of ER and is implicated in inducing DSBs at transcriptional enhancers regulated by ER. Using whole-genome sequencing in an engineered cell model lacking base excision repair (BER) function, we demonstrate that A3B preferentially targets transcriptionally active regulatory regions in an R-loop-dependent manner. Strand-specific DNA:RNA immunoprecipitation sequencing (ssDRIP-seq) and ssDNA-associated protein immunoprecipitation sequencing (SPI-seq) confirm that A3B binds to and deaminates ssDNA within R-loops, a process facilitated by ER transactivation. Furthermore, BER-mediated processing of A3B-induced uracil bases contributes to the formation of R-loop-associated DSBs, which are essential for ER-regulated gene activation. These findings establish a role for A3B in R-loop homeostasis and transcriptional regulation, with implications for understanding ER-driven genomic instability and potential therapeutic targeting of A3B.

Oestrogen (17-β E2; E2) plays a crucial role in the development and maintenance of the human reproductive system and secondary sexual characteristics. Prolonged exposure to E2 has been implicated in the pathogenesis of breast, ovarian, and uterine cancers[1]. E2 exerts its effects by binding and activating the oestrogen receptor (ER). Upon binding E2, ER translocates to the nucleus and acts as a transcription factor, recognising E2-responsive elements (EREs) in genomic DNA. This event triggers altered expression of numerous downstream genes[2]. The set of genes regulated by ER activation encompasses those encoding essential reproductive functions, including cell proliferation and survival, and dysregulated ER activity has been associated with cancer development[3].

In addition to its role in gene regulation, ER has been implicated in promoting cancer development by inducing DNA damage[4]. Importantly, short-term E2 stimulation in breast epithelial cells expressing ER leads to the formation of DNA double-strand breaks (DSBs)[5,6], contributing to chromosome instability and aneuploidy[7]. ER-positive breast cancers exhibit increased genomic damage, characterised by

[1]Centre for Cancer Drug Discovery, Division of Cancer Therapeutics, the Institute of Cancer Research, London, United Kingdom. [2]Shanghai Institute of Biological Products, Shanghai, China. [3]State Key Laboratory of Novel Vaccines for Emerging Infectious Diseases, China National Biotec Group Company Limited, Beijing, China. [4]Institute of Biomedical and Pharmaceutical Sciences, Guangdong University of Technology, Guangzhou, China. [5]GMU-GIBH Joint School of Life Sciences, Guangzhou Medical University, Guangzhou, Guangdong, China. [6]Key Laboratory of Synthetic Biology, Ministry of Agriculture and Rural Affairs, Advanced Synthetic Biology Institute at Hohhot, Agricultural Genomics Institute at Shenzhen, Chinese Academy of Agricultural Sciences, Shenzhen, China. [7]These authors contributed equally: Yu-jing Lu, Bingjie Chen. ✉e-mail: olivia.rossanese@icr.ac.uk; paul.workman@icr.ac.uk; paul.clarke@icr.ac.uk

mutations, alterations in gene copy number, and recombination[8,9]. ER-induced DNA damage is considered to be crucial for the activity of ER, since elements of the DNA repair pathway, such as transcription-coupled nucleotide excision repair (TC-NER) promote and participate in transcription activation[10,11]. The action of TC-NER also creates a favourable environment for transcription[12] by facilitating DNA demethylation[13], chromatin remodelling[14], and enhancer-associated RNA (eRNA) synthesis[15].

Various models have been proposed to elucidate the origin of oestrogen-induced DNA double-strand breaks (DSBs). These models include the involvement of DNA topoisomerase IIβ[16], the formation and processing of co-transcriptional R-loops[6], and DNA damage triggered and edited following deamination by apolipoprotein B mRNA-editing enzyme catalytic polypeptide-like 3B (APOBEC3B, A3B)[5]. A3B belongs to the APOBEC3 enzyme family, which comprises seven closely related DNA deaminases responsible for catalysing cytosine-to-uracil (C > U) editing of single-stranded DNA (ssDNA)[17] or RNA. Apart from their innate functions as immune defences against DNA viruses and transposons[18,19], several family members, including A3B, have been implicated in driving genetic heterogeneity in cancers by inducing C > T transitions and C > G transversions at 5′-TCW motifs (W = A or T)[8,20]. Elevated expression of A3B has been observed in more than 50% of primary breast cancers[20]. In addition, as the sole constitutively nuclear-localised member of the APOBEC3 family, A3B has emerged as one of the primary candidates underlying the APOBEC3 mutational signature identified in various cancers[20,21].

Furthermore, the activation of A3B has been shown to induce DNA damage repair (DDR)[9,22,23], where A3B directly interacts with ERα and edits ER binding sites, which are subsequently processed into DSBs by DNA base excision repair (BER) and non-homologous end joining (NHEJ) mechanisms[5]. The appropriate generation of these DSBs, along with subsequent epigenetic modifications, plays a critical role in determining the transactivation activity of ER[5]. Functional loss of A3B abolishes ER-driven cell proliferation and extends the therapeutic efficacy of tamoxifen in ER-positive xenograft models[24]. These findings underscore the role of A3B as an important regulator of ER activity.

Elevated levels of transcriptional activity result in the formation of chromatin-associated R-loop structures, characterised by the formation of a three-stranded structure comprising a DNA:RNA hybrid and a stretch of displaced ssDNA[25]. Previous research has indicated that the clearance of co-transcriptionally formed R-loops by the DDR machinery contributes to the generation of DSBs mediated by ER activation[6]. Given that the displaced ssDNA within R-loops could potentially serve as a substrate for A3B, it has been postulated that A3B may participate in R-loop-associated functions. This hypothesis gains support from studies utilising the APOBEC-Cas9 engineered system to edit cytosines in R-loop DNA[26], a recent investigation revealing the influence of A3B on R-loop homoeostasis and consequent R-loop-associated mutagenesis[27]. However, despite these endeavours, it remains unknown whether the editing of R-loops by A3B is directly associated with its role as an ER coactivator in cells.

To comprehensively investigate the impact of A3B-mediated R-loop editing on the co-activation of ER-regulated transcription, we used genome-wide profiling of A3B editing sites in ER-dependent models. This approach offers insights into potential hotspots of A3B editing, shedding light on the underlying mechanism governing A3B-dependent DNA deamination and its regulatory role in ER activity. Previous large-scale whole-genome sequencing (WGS) studies conducted on cancer cell lines and primary human tumour samples have successfully identified APOBEC editing sites[9,28]. However, these investigations are potentially influenced by a survivorship bias, as the detection of A3B-edited sites is restricted to those that evade repair by BER, which rectifies U:G mispairing resulting from A3B editing[29,30]. This limitation will also impair studies employing A3B-overexpressing cell models[31–34]. To overcome this challenge, we employed an engineered BER-deficient breast cancer cell model to sample A3B editing sites and adopted a multi-omic approach to elucidate the intricate role of A3B in the molecular mechanism governing ER-regulated gene expression.

## Results

### Detection of A3B DNA deamination sites using a BER-deficient cancer cell model

To identify A3B deamination sites with potential functional impacts on protein coding or gene regulation, we conducted WGS of ER-positive T47D human breast cancer cells expressing doxycycline-inducible A3B. As a control, we adopted an enzymatically inactive A3B E68Q/E255Q double mutant (A3B**), which was confirmed to lack detectable enzymatic activity in multiple cell line models (Fig. 1a and Supplementary Fig. 1). Bicistronic expression of the humanised bacteriophage PBS2 uracil glycosylase inhibitor (hUGI) peptide was employed to inhibit BER-associated uracil-DNA glycosylase (UDG/UNG)[20,30]. We showed that lentiviral hUGI expression prevents the removal of A3B-induced uracil bases, allows more effective capture of A3B editing events, and it is sufficient to block the majority of UDG/UNG activity within cells (Supplementary Fig. 2). The selection of the T47D cell line was based on its TP53 loss-of-function mutation, which is known to confer protection against synthetic lethality induced by ectopic A3B expression[35].

Doxycycline-induced expression of A3B/hUGI and effective uridine incorporation were confirmed by immunoblotting and slot blotting using *Mycobacterium smegmatis* UdgX as a uridine DNA sensor (Fig. 1a and Supplementary Fig. 3a, b). The cells were then exposed to doxycycline for five days, and genomic DNA samples were sequenced from five individual T47D clones expressing A3B-hUGI (A + colonies, Fig. 1b). For control purposes, sequencing results from three uninduced colonies (A- colonies) were included to represent single-nucleotide variants (SNVs) arising from isolating individual clones. In addition, three T47D colonies co-expressing hUGI and the enzymatically inactive A3B** (M + colonies) were also included. A3B** neither interfered with the activity of endogenous A3B (Supplementary Figs. 1, 2) nor triggered an unfolded protein response (Supplementary Fig. 4), indicating that it has no functional impact on the endogenous A3B enzyme. The M + colonies were therefore expected to capture base-editing events mediated by endogenous APOBEC/AID family enzymes in T47D cells, predominantly A3B[5]. Slot blots with UdgX uridine DNA sensor confirmed that both the endogenous and induced wild-type A3B in the individual colonies retained full enzymatic activity (Supplementary Fig. 3c, d).

Joint mutation calling for the separated colonies was performed using a parental T47D reference sample. Analysis of the WGS data confirmed that the integrated lentiviral construct, including the A3B open-reading frame and flanking vector sequence, was sequence-intact, with no deep or non-synonymous mutations detected following A3B induction (Supplementary Table 1). Subsequent variant analysis revealed a significantly higher proportion of non-CpG C > T mutations in the A + colonies compared to the M + and A- colonies (Fig. 1c). To better characterise the nature and spatial distribution of these mutations, we analysed whether they occurred individually or as clusters. As indicated by previous studies, mutations can occur as clusters with various patterns, including large foci of close-together mutations also described as kataegis (thunderstorm) or alternatively, clusters of 3-4 mutations that occur in short distances known as omikli (fog)[22]. Studies using BER-functional models have detected prevalent kataegis-like clusters of mutations associated with increased A3B expression[31,33]. Here, we analysed intermutational distance for the detected mutations (Fig. 1d and Supplementary Fig. 5), and employed the HyperClust algorithm for the detection of localised somatic hypermutations[22]. Figure 1e clearly shows that the A+ colonies exhibit a greater number of non-recurrent and diffused clusters of mutations consistent with omikli rather than the previously reported kataegis. This observation suggests that the inhibition of BER in our cell model captures a distinct

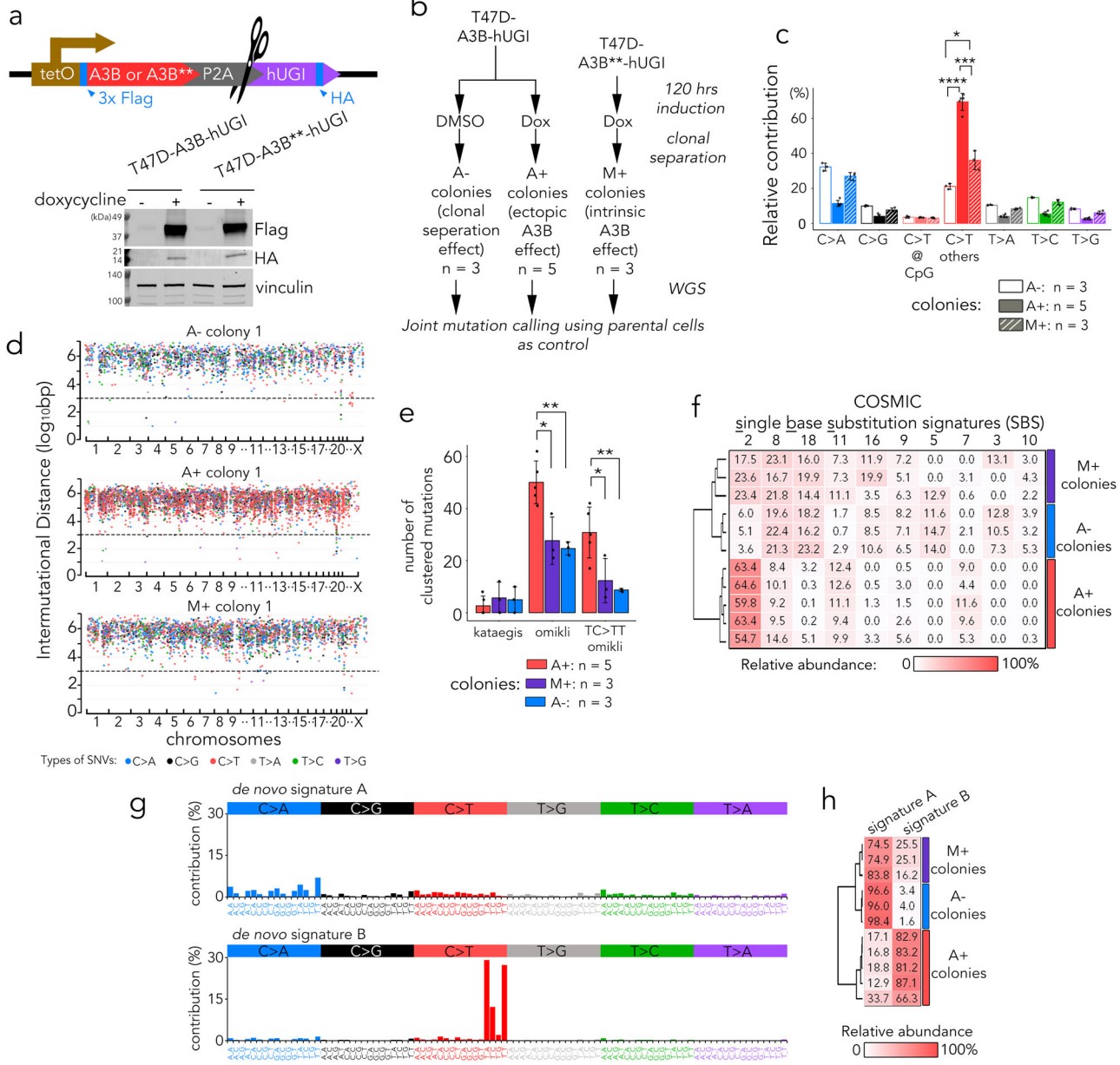

**Fig. 1 | Detection of A3B deamination sites in BER-deficient T47D cell models.** **a** Lentiviral inducible system used in this study and immunoblotting of human T47D breast cancer cells with and without 24 h induction by 1 μg/ml doxycycline. A3B** denotes A3B with E68Q/E255Q mutations. The immunoblot shown is representative of two independent experiments. **b** Schematic of sample preparation for WGS studies. **c** Mean relative contribution (%) of the indicated types of point mutations in the three indicated sample groups. Bars represent means; error bars, SD. Statistical significance was assessed by one-way ANOVA across groups (* $p < 0.05$; *** $p < 10^{-3}$; **** $p < 10^{-4}$). For non-CpG C > T mutations, $p = 1.6 \times 10^{-5}$ for A+ vs A−, $p = 3.7 \times 10^{-4}$ for A+ vs M+, and $p = 0.0102$ for M+ vs A−. For A− and M+ colonies, three independent colonies were sampled; for A+ colonies, five independent colonies were sampled. **d** Waterfall plot of intermutational distance (IMD) for each mutation identified in representative colonies. Dotted line denotes IMD ≤ 10³ bp.

**e** Mutation clusters detected by HyperClust for three sample groups. Bars represent mean values; error bars, SD. Statistical significance was assessed by one-way ANOVA across groups (*$p < 0.05$; **$p < 0.01$). For omikli mutations, $p = 1.9 \times 10^{-2}$ for A+ vs M+ and $p = 7.2 \times 10^{-3}$ for A+ vs A−. For omikli mutations at TC > TT sites, $p = 1.5 \times 10^{-2}$ for A+ vs M+ and $p = 6.0 \times 10^{-3}$ for A+ vs A−. For A− and M+ colonies, three independent colonies were sampled, and for A+colonies, five independent colonies were sampled. **f** Heat map showing the cosine similarity scores of top-ranking COSMIC signatures for each indicated sample. **g** Relative contribution of each indicated trinucleotide alternation to the two top-ranking de novo mutational signatures identified by NMF analysis. **h** Heat map showing the cosine similarity scores of mutational signatures in (G). for each indicated groups of samples. Source data are provided as a Source Data file.

pattern of A3B-mediated mutations in alignment with deficient DNA mismatch repair frequently observed in human cancer genomes[22].

Analysis of the resulting mutations against predefined COSMIC single-base substitution (SBS) signatures using non-negative matrix factorisation (NMF)[28] identified enrichment of the SBS2 signature, attributed to the activities of the AID/APOBEC family of cytidine deaminases (Fig. 1f). De novo NMF signature extraction revealed two

prevalent signatures among the sequenced control colonies: signature A, more prevalent in the control A- or M+ colonies, potentially accounting for the variation among T47D subclones, and signature B, primarily enriched in A + colonies, characterised by the A3B-preferred 3′-TCW trinucleotide motif resembling the mutational process associated with A3B editing (Fig. 1g, h). In addition, analysis of the M+ colonies revealed a measurable but weaker enrichment of C > T

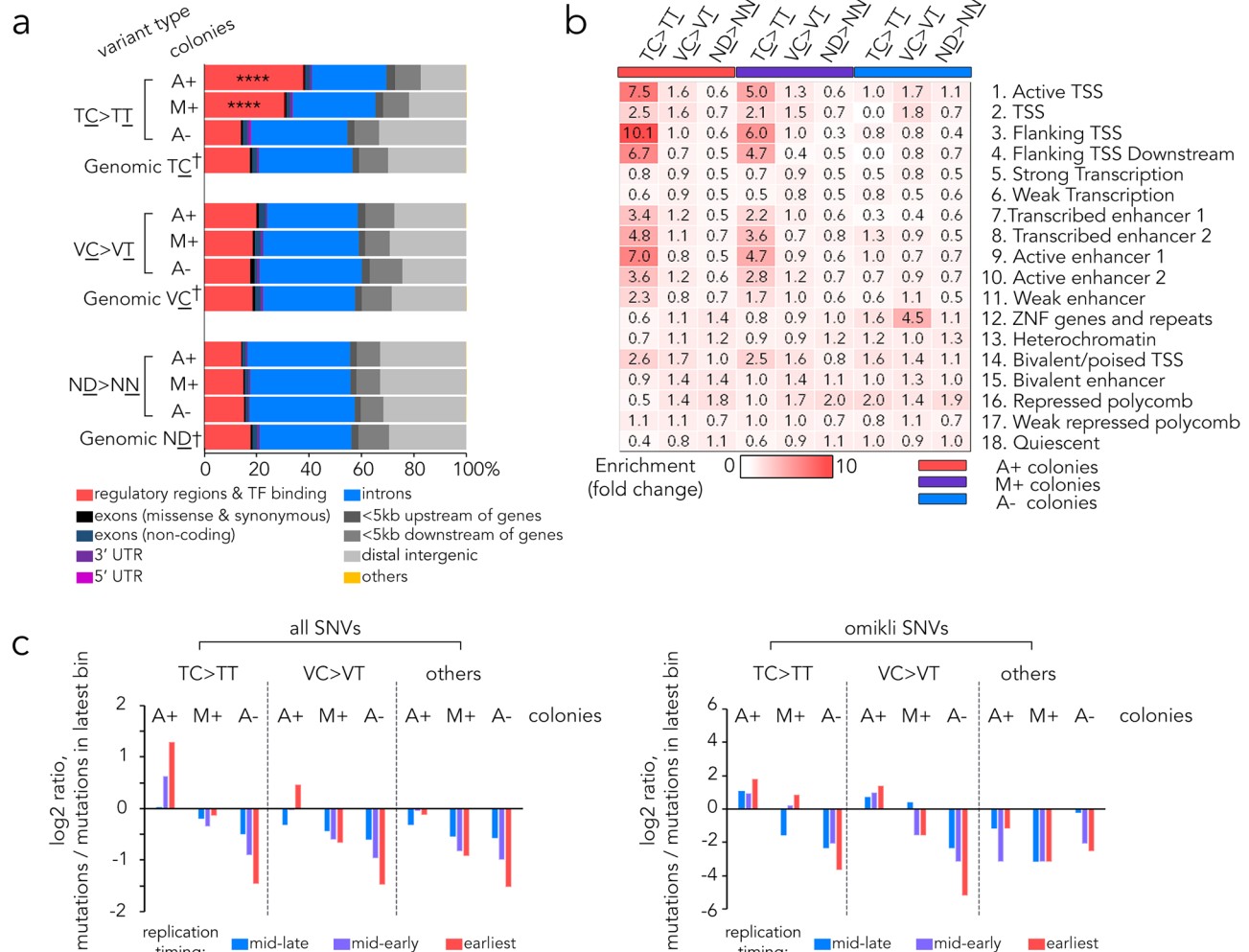

**Fig. 2 | A3B preferentially deaminates genomic regulatory regions.**
**a** Distribution of variant consequences as determined by VEP for indicated types of variants in T47D cells. †: random sampling of $10^6$ sites. ****: $p < 10^{-4}$, $\chi^2$ test (two-sided) comparing to random sites. **b** Heat map showing enrichment score of chromatin states for ± 50 bp regions flanking point mutations using ChromHMM model depicted in Supplementary Fig. 2. **c** Rates of indicated type of mutations in DNA replication timing quartiles, relative to the latest replicating quartile, for SNVs from indicated sample groups. Replication timing quartiles were derived using Repli-seq of T47D cells (ENCODE: ENCFF440QFG). Source data are provided as a Source Data file.

substitutions at the A3B-preferred 3′-TCW trinucleotide motif and partial retention of SBS2 and Signature B. This effect, although attenuated compared with wild-type A3B induction, reflects deamination by the endogenous A3B enzyme whose activity becomes unmasked when BER is suppressed by the bi-cistronically expressed hUGI peptide. These data demonstrated that the BER-deficient breast cancer model captures A3B activity on genomic DNA in the absence of interference by downstream DNA repair mechanisms.

## A3B preferentially deaminates genomic regulatory regions

We investigated the functional consequences of the captured A3B-editing sites using the Ensembl Variant Effect Predictor (VEP)[36]. Compared with random genomic locations, we observed a significant enrichment of TC > TT substitutions in A + colonies within regulatory and transcription factor (TF) binding regions, but not in protein-coding regions. Intrinsic APOBEC-mediated TC > TT mutations in M + colonies followed a similar but less significant trend, while TC > TT mutations arising from colony separation (A- colonies) displayed no obvious preference. Also, this enrichment was not observed for control VC > VT or ND > NN substitutions (Fig. 2a). To better understand the functional roles of these transcriptional regulatory regions, we constructed an 18-chromatin state ChromHMM model for the parent

T47D cells, utilising previously published ChIP-seq data (Supplementary Fig. 6). We then examined the overlap between the captured mutations and specific chromatin states derived from this model using methods described previously[37] (Fig. 2b). In the A+ colonies, we identified a significant overrepresentation of TC > TT mutations at transcription start sites (TSS) and enhancers, rather than main body (UTR, exon or intron) of the gene, particularly in the flanking regions of TSS and active enhancers. A similar pattern, although to a lesser extent, was observed for intrinsic APOBEC-induced mutations in M + colonies. In addition, we analysed DNA replication timing using parent T47D Repli-seq data from the ENCODE dataset (ENCFF440QFG), which showed that the mapped TC > TT substitutions in A + colonies, either as SNV or omikli events, were predominantly associated with early DNA replication sites (Fig. 2c). These findings align with previous observations indicating that A3B targets actively transcribed genomic regions and those involved in early replication[22], and underscore the potential functional significance of A3B-associated modifications[22]. Overall, our data provide evidence of a direct association between A3B-mediated base-editing and sites of transcriptional regulation. These results are consistent with the previously established role of A3B[5], and argue against a primary role for A3B's DNA deamination activity in altering protein-coding sequences.

## A3B binds and deaminates ssDNA in R-loops

Next, we sought to understand the sequence context flanking A3B-mediated mutations to determine features that may facilitate substrate recognition by A3B. In A+ colonies, we observed that TC > TT substitutions are accompanied by a pronounced GC-skewness in the flanking DNA sequences, indicating an overrepresentation of guanine bases in the same DNA strand as the TC > TT editing event (Fig. 3a). Furthermore, there was an enrichment of G-quadruplex-forming sequences within a ± 500 bp region surrounding A3B-editing sites (Fig. 3a). A similar, albeit weaker, effect was observed around TC > TT substitutions in M + colonies, which represents the deamination by intrinsic A3B. In contrast, no such patterns were observed for TC > TT or other di-nucleotide substitutions derived from A- colonies (Fig. 3a and Supplementary Fig. 7). Both GC-skewness and G-quadruplex-forming sequences are known to facilitate the formation or stabilisation of R-loops[38]. Therefore, we explored the potential role of R-loops as substrates for A3B activity by first verifying their presence at A3B editing sites. We employed strand-specific DNA:RNA immunoprecipitation sequencing (ssDRIP-seq) combined with RNase H-dependency to corroborate mapping DNA:RNA hybrids present in R-loops, GC-skewness, and MNase-seq analyses in parent T47D cells (Supplementary Fig. 8). ssDRIP-seq specifically captures and maps DNA:RNA hybrid structures, enabling genome-wide identification of R-loops[39]. The results identified significantly elevated R-loop signals near TC > TT substitutions in A + colonies. Strand-specific sequencing revealed that the TT > TC substitutions occurred on the displaced single-stranded DNA (ssDNA) strand of the R-loop, consistent with A3B's substrate preference for ssDNA (Fig. 3b). A similar but less pronounced enrichment of R-loop signals was found at TC > TT substitutions in M + colonies, while no such signal was found in other di-nucleotide substitutions (Supplementary Fig. 9), confirming a similar site preference for intrinsic A3B base editing.

To further investigate the correlation between R-loops and A3B-mediated editing, we conducted immunoprecipitation experiments using the R-loop-binding S9.6 antibody, a well-established method for isolating R-loops and their associated proteins[40]. Similar to Top1, a validated R-loop binding protein[40], A3B was found to co-immunoprecipitate with R-loops. This interaction was disrupted upon exogenous expression of RNaseH1, an enzyme that degrades RNA:DNA hybrids and removes R-loop structures (Fig. 3c).

To map the genomic occupancy of flag-tagged A3B and test its dependency on R-loops, we employed ssDNA-associated protein immunoprecipitation sequencing (SPI-seq) to capture A3B-bound ssDNA[41] in Flag-A3B-expressing T47D cells, with or without co-expression of RNaseH1 to eliminate R-loop formation. We selected SPI-seq as this method would capture A3B-bound ssDNA in addition to dsDNA, the latter typically identified by conventional ChIP-seq. Following quality control analysis and validation of SPI-seq specificity by qPCR using T47D samples with or without A3B depletion (Supplementary Fig. 10a–c), we found the overexpression of RNaseH1 greatly reduced the number of Flag-A3B peaks captured and hindered Flag-A3B's binding to both promoter and enhancer regions (Supplementary Fig. 10b–f). A close inspection of the Flag-A3B peaks eliminated by RNaseH1 co-expression revealed enrichment of R-loop sequences and G-quadruplex motifs relative to the unaffected peaks (Fig. 3d). Notably, SPI-seq captured A3B-bound ssDNA fragments were found to be localised within the body of R-loops, the signal abundance of which can be affected by RNaseH1 expression (Fig. 3e). We also observed a more pronounced decrease in A3B SPI-seq peaks signal in the neighbourhood of R-loops upon RNaseH1 expression (Fig. 3f), indicating that A3B binding sites proximal to R-loops are more susceptible to R-loop clearance, thus demonstrating a dependency of A3B's ssDNA binding on R-loop formation. In addition, enrichment of the A3B binding signal can be observed in previously identified A3B-driven TC > TT substitutions, the strength of which was attenuated upon R-loop clearance by RNaseH1 overexpression, suggesting C > T DNA editing by A3B can also be affected by R-loop availability in cells (Supplementary Fig. 11). Collectively, our ssDRIP-seq and SPI-seq data support the notion that R-loop formation facilitates A3B binding and cytidine deamination of genomic DNA.

## ER-induced R-loop formation facilitates A3B binding

Given that the activation of ER triggers the co-transcriptional formation of R-loops, which can influence gene expression[6], we postulated that increased levels of genomic R-loops would enhance the availability of substrates for A3B and subsequently facilitate A3B binding to ER-activating enhancer regions. To investigate this, we employed SPI-seq and ssDRIP-seq to map the binding sites of flag-tagged A3B and R-loops responsive to E2 stimulation in T47D cells, enabling the examination of their spatial-temporal relationship. Here, we employed quantitative SPI-seq (with $n = 2$ replicates) to identify genomic regions bound by Flag-tagged A3B and quantitative ssDRIP-seq (with $n = 3$ replicates) to map R-loop formation in T47D cells treated with or without E2 (Fig. 4a and Supplementary Fig. 12), allowing us to analyse their spatial-temporal relationship. Consistent with previous findings[5,6], we observed a global increase in A3B binding (23,711 peaks; 53% of all peaks) and R-loop formation (18,567 peaks; 16% of all peaks) after two hours of E2 exposure (Fig. 4b). A closer inspection found that A3B binding sites were enriched for promoter and enhancer sites irrespective of E2 treatment (Supplementary Fig. 13). In contrast, only R-loops induced by E2 treatment showed a trend of enrichment near TSS regions, whereas R-loops formed independently of E2 did not (Supplementary Fig. 14). Remarkably, a substantial portion of newly-formed E2-dependent A3B binding sites were found in proximity to R-loops following E2 exposure ($p < 10^{-5}$, $\chi^2$ test), and correspondingly, R-loops near A3B binding sites exhibited increased sensitivity to E2 induction (Fig. 4b). A substantial overlap was seen between E2-induced A3B binding sites and R-loops (Fig. 4c), as well as a significant enrichment of E2-induced R-loops in proximity to E2-induced A3B binding sites ($p < 10^{-5}$, $\chi^2$ test, Fig. 4d). These observations strongly suggest a link between R-loop dynamics and A3B's substrate binding following E2 treatment.

Our data suggest a potential causal relationship underlying R-loop formation and A3B binding at sites of E2-induced gene expression. However, depletion of A3B using siRNA did not impact the overall levels of R-loop signal or the increase in R-loops upon E2 induction determined by S9.6 antibody-based slot blotting (Supplementary Fig. 15). Furthermore, quantitative ssDRIP-seq analysis revealed an increase in R-loops proximal to A3B binding sites despite A3B knockdown (Supplementary Fig. 16a). By employing the DESeq2 quantification method featuring two-factor interaction analysis[42], we quantitatively examined the combinatorial effects of A3B siRNA treatment and E2 induction on R-loop formation (Supplementary Fig. 16b), which showed minimal interaction between these two factors, indicating that A3B depletion is unlikely to directly affect E2-induced R-loop formation. On the other hand, more than half of the E2-induced A3B binding sites were found either located near (<2 kb) E2-induced R-loops (34%) or pre-existing R-loops prior to E2 treatment (28%, Fig. 4e), with R-loop-proximal A3B peaks displaying a higher degree of induction upon E2 treatment compared to R-loop-distal A3B peaks (Supplementary Fig. 16c).

These findings suggest that elevated A3B binding is unlikely to drive the induction or formation of R-loops, but instead, the availability of R-loops may dictate E2-induced A3B binding and editing chromatin in or adjacent to the R-loop. To further examine this, we investigated the impact of RNaseH1 overexpression on A3B binding in seven loci characterised by well-defined ER binding and R-loop formation (Fig. 4f). In contrast to the two A3B binding sites lacking R-loops at promoters of *TNFSF8* and *RMST*, clearance of R-loops by RNaseH1 overexpression impaired E2-induced A3B binding (Fig. 4g

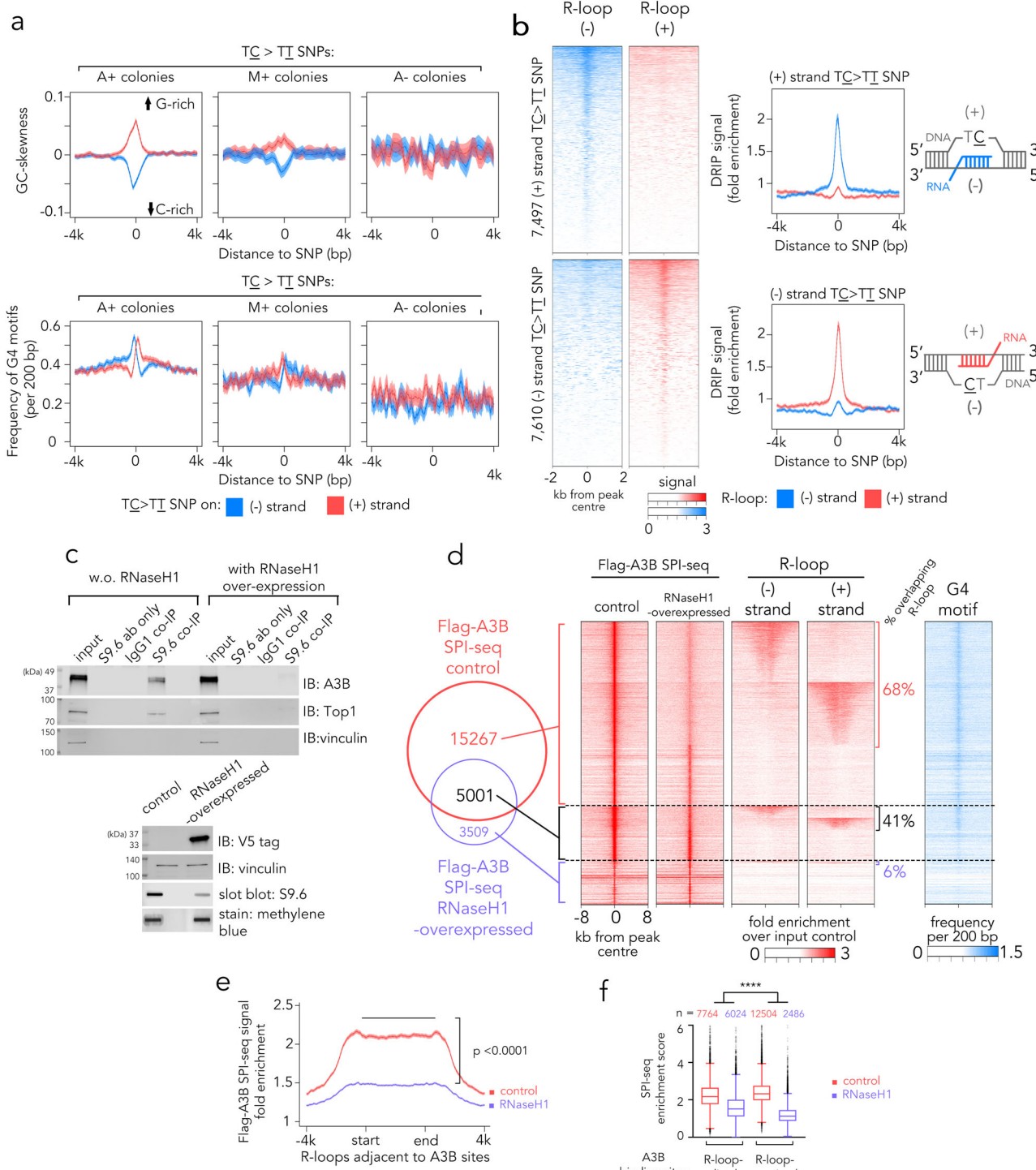

**Fig. 3 | A3B binds and edits R-loops. a** Profiles of GC skewness and frequency of G-quadruplex (G4) motifs in regions flanking the TC > TT SNP. The line denotes the average value, and the shaded areas 95% CI. **b** Heat maps showing T47D cells' ssDRIP-seq signals in regions flanking the TC > TT SNP identified in A + colonies (left), their corresponding signal profile plots (middle), and schematics demonstrating the position of R-loop in relation to the site of TC > TT SNPs (right). For signal profile plots, the line denotes the average value and the shaded areas 95% CI. **c** S9.6 co-immunoprecipitation assay of T47D cells transfected with or without RNaseH1-encoding vectors for 24 h (top), and corresponding immunoblots and slot blots of input samples (bottom). Results are representative of two independent experiments with similar results. **d** Heat maps of enrichment signal from indicated

sequencing experiments at the flanking regions of Flag-A3B SPI-seq peaks in T47D cells. **e** Profile of Flag-A3B SPI-seq signal for R-loops that are proximal to A3B sites in T47D cells. The line denotes the average value and the shaded areas 95% CI. The statistic represents one-way ANOVA using enrichment scores at the indicated regions. **f** Tukey boxplots showing enrichment scores from T47D cells' SPI-seq experiments. Boxes indicate the interquartile (25th-75th) range (IQR), the centre line indicates the median, and whiskers extend to the most extreme data points within 1.5 × IQR; points beyond the whiskers represent outliers. The number of SPI-seq peaks in each group is shown above the boxes. ****: $p \leq 10^{-4}$, two-way ANOVA assessing the effect of R-loop proximity on response to 24 h RNaseH1 over-expression $p = 2.2 \times 10^{-16}$. Source data are provided as a Source Data file.

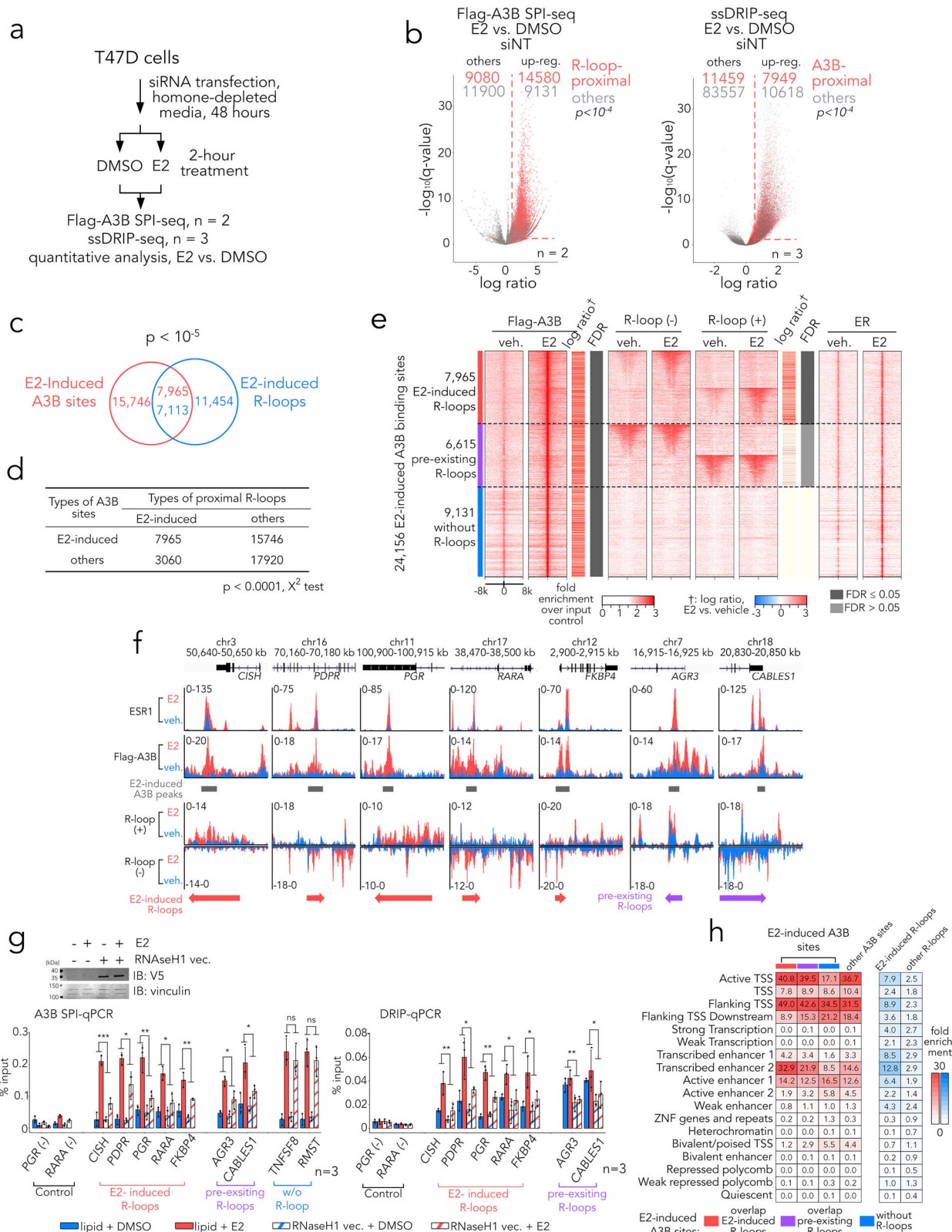

and Supplementary Fig. 17) in T47D and MCF7 cells. These results confirm that the binding of A3B to specific regions induced by E2 exposure involves R-loop serving as A3B substrates.

Our findings suggest a model where the formation of R-loops remodels the genomic landscape surrounding EREs, facilitating the recruitment of A3B. Using our ChromHMM model, we identified that E2-induced A3B binding sites overlapping with R-loops exhibit high enrichment scores in regions proximal to transcription start sites (TSS), TSS flanking regions and transcribed enhancers (Fig. 4h). Importantly, we found that E2-induced A3B binding sites showed a distinct preference towards transcriptional enhancer sites undergoing active transcription, which was not observed at other A3B binding sites. This coincides with the finding that E2-induced R-loops display the highest enrichment in actively transcribed enhancer regions. In

**Fig. 4 | R-loops induced by ER activation facilitate A3B binding. a** Schematic of the quantitative A3B SPI-seq (two biological replicates per condition) and ssDRIP-seq (three biological replicates per condition) experiments. **b** Effect of 2 h E2 stimulation on Flag-A3B binding sites (SPI-seq, left) and R-loop sites (ssDRIP-seq, right) in T47D cells. R-loop-proximal Flag-A3B sites are in red. Upregulated hits (FDR ≤ 0.05, fold change ≥ 1.5) are indicated by red dotted lines. *P*-values from two-sided $\chi^2$ tests report the non-random association between R-loop or A3B proximity and E2 response. **c** Overlap between E2-induced Flag-A3B binding sites and E2-induced R-loops in T47D cells. *P*-value from two-sided $\chi^2$ test for independence relative to the genomic background. **d** Contingency table showing distribution of various types of A3B sites and the result from $\chi^2$ analysis (two-sided). **e** Heat maps showing Flag-A3B SPI-seq, ssDRIP-seq and ER ChIP-seq signals in ± 8 kb regions flanking E2-induced Flag-A3B binding sites in T47D cells. **f** Representative signal tracks for ER ChIP-seq, Flag-A3B SPI-seq and ssDRIP-seq at E2-induced Flag-A3B

binding sites that are proximal to E2-induced R-loops (*CISH, PDPR, PGR, RARA* and *FKBP4*) or to pre-existing R-loops (*AGR3* and *CABLES1*) in T47D cells. Data represent fold change in signal (reads per million, CPM) over input. **g** Bar graphs showing Flag-A3B SPI-qPCR and DRIP-qPCR in T47D cells transfected with or without RNaseH1 for 24 h before 2 h stimulation with DMSO or 100 nM E2. Data are mean of three biological replicates; error bars, SD. Two-way ANOVA *P*-values for the effect of RNaseH1 on the E2 response are: A3B SPI-qPCR, *CISH* $1.1 \times 10^{-4}$, *PDPR* 0.014, *PGR* $2.4 \times 10^{-3}$, *RARA* $7.4 \times 10^{-3}$, *FKBP4* $2.3 \times 10^{-3}$, *AGR3* 0.015, *CABLES1* 0.035; DRIP-qPCR, *CISH* $2.6 \times 10^{-3}$, *PDPR* 0.013, *PGR* $3.4 \times 10^{-3}$, *RARA* $6.4 \times 10^{-3}$, *FKBP4* 0.028, *AGR3* $8.3 \times 10^{-3}$, *CABLES1* 0.0158. *$P \leq 0.05$; **$P \leq 0.01$; ***$P \leq 0.001$; ns, $P > 0.05$. **h** Heat map showing enrichment scores for ChromHMM chromatin states at indicated A3B binding sites or R-loops in T47D cells. Source data are provided as a Source Data file.

addition, analysis of transcription factor (TF) binding DNA motifs surrounding E2-associated A3B binding sites reveals a distinct set of TFs that may be enriched in E2-regulated R-loop proximal peaks, particularly characterised by G-rich sequence motifs, in contrast to R-loop-distal A3B peaks (Supplementary Fig. 16d). This observation further supports the notion that formation of R-loops, facilitated by G-rich motifs, may actively reshape the localisation of the A3B binding site. Overall, our data strongly suggest that the increased availability of R-loops induced by ER transactivation could regulate A3B binding.

## A3B promotes DSB formation at E2-induced R-loops

Several investigations have demonstrated that the formation of double-strand breaks (DSBs) induced by ER transactivation can result from the processing of co-transcriptionally formed R-loops or A3B editing sites through DDR mechanisms[5,6]. However, whether these two processes are functionally linked remains unclear. To explore the interconnection between these processes, we employed DSBCapture-seq to accurately map genomic DSB sites[43] and investigate the dependence of E2-induced DSBs on A3B binding and editing and/or R-loop formation. T47D control and A3B-depleted cells were exposed to DMSO or E2 for a duration of two hours, followed by DSBCapture-seq analysis (Fig. 5a). Differential peak analysis of the sequencing outcomes using DiffBind[44] revealed two categories of DSBs: E2-induced DSBs and A3B-modified DSBs (Fig. 5a). Our results corroborate the previously reported extensive DNA damage triggered by ER activation[4–6,45] and demonstrate that the 15,259 E2-induced DSB peaks account for 30% of all detectable DSBs in the basal parent T47D cell-line genome. 5308 DSB peaks were modified by A3B depletion, with the majority significantly (88%, $p < 10^5$ by $X^2$ test) overlapping with the E2-induced DSBs (Fig. 5b). This enrichment of A3B-modified DSBs within the E2-induced DSB set aligns with previous reports [5]. Support for the involvement of A3B in these DSBs came from our evidence that nearly all (98.6%) of the A3B-modified DSB sites we detected overlapped with the E2-induced A3B binding sites we identified through quantitative SPI-seq (Fig. 5c). However, A3B binding alone was not sufficient to predict A3B-dependent DSB formation, as 74% of the E2-induced A3B-independent DSBs also overlapped with A3B binding sites, suggesting there was an additional determinant for A3B dependency. We also observed a prerequisite for R-loop formation preceding A3B binding in the generation of A3B-modified DSBs, since a significant overlap ($p < 10^5$ by $X^2$ test) with R-loops was detected for these A3B-dependent DSBs compared to the A3B-independent E2-induced DSBs (Fig. 5c). Accordingly, both R-loop formation and A3B binding served as distinguishing factors between A3B-modified DSBs and A3B-independent DSBs (96% vs. 42% of all DSBs, Fig. 5d) and were reliable indicators of the impact of A3B depletion on DSB formation (Fig. 5e, f). Thus, our findings support the involvement of R-loop formation in the generation of A3B-induced DSBs.

Upon closer examination of the regions flanking DSBs, distinctive features associated with A3B-dependent DSBs emerged, compared

with those at A3B-independent sites. These features included heightened bidirectional transcription activities (determined by global run-on sequencing, GRO-seq), increased occupancy of RNA polymerase II (determined by PolII ChIP-seq), and the presence of denser G-quadruplex-forming sequences (determined by G4Hunter sequence analysis), all of which promote R-loop formation[46] (Fig. 5g). Utilising genome-wide analysis with our ChromHMM model, we observed that A3B-dependent DSBs exhibited greater enrichment at active or flanking transcription start sites (TSS) and transcriptionally active enhancers (Supplementary Fig. 18). We also found these same chromatin states, characterised by transcriptional activity, were enriched for E2-induced R-loops, suggesting a reliance on R-loops during the conversion of A3B editing sites into DSBs (Fig. 4h).

The observed association between A3B binding and transcriptionally active R-loop regions raised the possibility that the formation of A3B-dependent DSBs involves mechanisms linked to active transcription, such as TC-NER. The increased transcriptional activity and RNA polymerase II occupancy at A3B-dependent DSBs could thus provide substrates and conditions favourable for TC-NER, which has been implicated in R-loop resolution and subsequent generation of single-strand breaks (SSBs)[6]. To investigate this further, we analysed seven A3B binding sites that overlapped with R-loops and were associated with A3B-dependent DSBs (Fig. 5h). We examined the impact of RNAi-mediated depletion of Cockayne syndrome group B (CSB) protein and XPG/XPF endonucleases, which are components of TC-NER required for the conversion of R-loops into DNA strand breaks[47]. Using quantitative PCR analysis of DSBCapture-enriched DNA samples, we discovered that depletion of these TC-NER components was sufficient to reduce the formation of E2-responsive A3B-dependent DSBs in T47D and MCF7 cells (Fig. 5i, Supplementary Fig. 19). Single depletion of either XPG or XPF caused only a slight reduction in E2-induced A3B-dependent DSB formation, whereas combined depletion almost completely abolished DSB production at all examined loci (Supplementary Fig. 20). Because XPF and XPG can incise either strand of the R-loop without strict strand preference, the modest effect of single depletion suggests functional redundancy between these endonucleases, each being individually capable of mediating the incision events required for R-loop processing and subsequent DSB formation. These findings demonstrate that A3B-driven DSB formation requires cooperative activity with the TC-NER machinery.

## Blocking the processing of A3B editing sites into DSBs impairs E2 response

The genomic localisation of A3B binding sites, particularly those in proximity to R-loops and associated with double-strand break (DSB) formation, implies a regulatory role for the downstream processing of A3B-mediated base editing. To examine the functional impact of blocking this processing step, we performed RNA-seq analyses using T47D cells containing the doxycycline-inducible, catalytically inactive A3B** mutant expressed together with hUGI (Fig. 6a). As shown

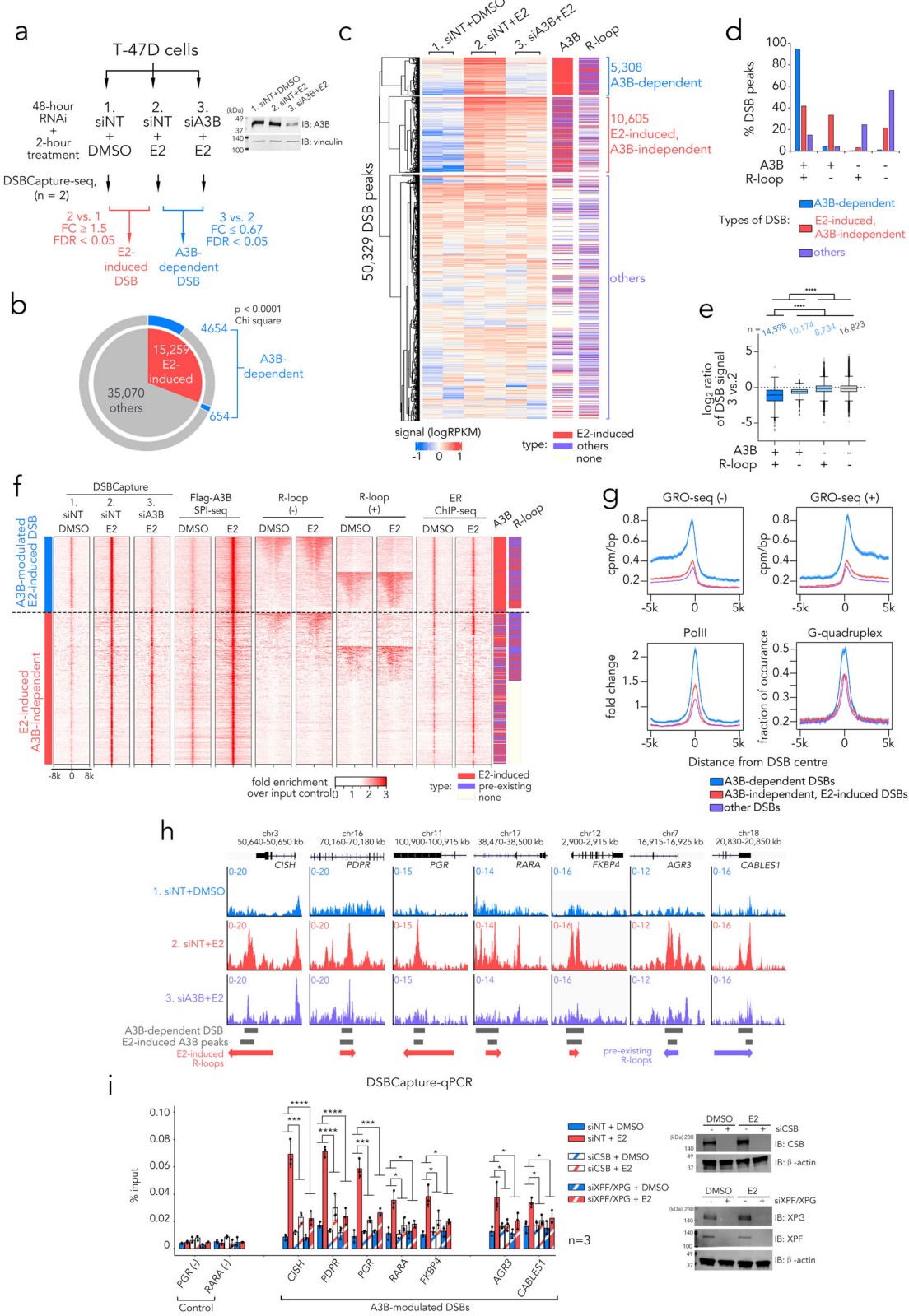

previously (Fig. 1 and Supplementary Figs. 1–4), A3B** has no detectable enzymatic activity and does not interfere with endogenous A3B, thereby serving as a non-catalytic control that allows assessment of uracil repair-dependent processes. By co-expressing hUGI, we effectively blocked the conversion of A3B-induced uracil lesions into repair intermediates that give rise to DSBs (Supplementary Fig. 2). Following exposure to E2 for two hours, we collected RNA samples from cells

with or without doxycycline induction of A3B** and performed RNA-seq. To assess differential gene expression, we employed the interaction modelling approach using the DESeq2 programme to examine the impact of loss of endogenous A3B activity on the E2 response[42]. Our results demonstrated that the expression of A3B**-hUGI dampened the E2-regulated response of 87% of genes (Fig. 6b), with 90 genes showing statistically significant interaction models based on DESeq2 analysis

**Fig. 5 | A3B promotes E2-induced DSB formation at R-loops. a** Schematic of quantitative DSBCapture in T47D cells (two biological replicates per condition). **b** Pie chart summarising DSBCapture results; *P*-value from two-sided $\chi^2$ test for non-random association between A3B dependence and E2-induced DSBs. **c** Heat maps of normalised DSB signals in T47D cells; each row is one DSB site, and each column a sample, with overlapping A3B/R-loop features indicated. **d** Bar graph showing percentage of DSB peaks overlapping A3B and/or R-loop. **e** Tukey box-plots of $\log_2$ DSBCapture signal ratio between A3B and non-targeting siRNA-transfected cells after 2 h E2. Boxes show IQR, centre line the median, whiskers $1.5 \times$ IQR, points beyond whiskers are outliers; DSB peak numbers are shown above boxes. Two-way ANOVA; **** $p \leq 10^{-4}$. **f** Heat maps for DSBCapture, A3B SPI-seq, ssDRIP-seq and ER ChIP-seq signals flanking $\pm 8$ kb of the centre of DSBs in T47D cells. Also shown are bars indicating types of overlapping Flag-A3B SPI-seq and ssDRIP-seq peaks. **g** Profiles of GRO-seq, Pol II ChIP-seq (T47D) and G-quadruplex frequency around the indicated DSB classes (as in **c**). Lines show the mean signal or frequency; shaded areas show 95% CI. **h** Representative signal tracks for DSBCapture at E2-induced Flag-A3B binding sites that are proximal to E2-induced R-loops in T47D cells. Data represents fold change in CPM over the input control. **i** Bar graphs showing DSBCapture-qPCR in T47D cells treated with NT, CSB, or XPF/XPG siRNA for 48 h before 2 h stimulation with DMSO or 100 nM E2. Data represent the mean of three biological replicates ± SD. Two-way ANOVA *P*-values for the effect of CSB knockdown on the E2 response are: *CISH* $1.3 \times 10^{-4}$, *PDPR* $5.863 \times 10^{-5}$, *PGR* $1.099 \times 10^{-4}$, *RARA* 0.011, *FKBP4* 0.033, *AGR3* 0.04, *CABLES1* 0.026. Two-way ANOVA P values for the effect of XPF/XPG knockdown are: *CISH* $8.609 \times 10^{-5}$, *PDPR* $5.863 \times 10^{-5}$, *PGR* $1.099 \times 10^{-4}$, *RARA* 0.011, *FKBP4* 0.013, *AGR3* 0.031, *CABLES1* 0.041. Source data are provided as a Source Data file.

(Fig. 6c). Gene set enrichment analysis (GSEA)[48] using the MSigDB hallmark genes[49] identified a significant overlap between the 90 E2-responsive genes whose regulation is modulated by A3B-hUGI expression and those regulated by oestrogen response pathways (Fig. 6d), suggesting the involvement of uracil repair of A3B-mediated DNA deamination sites during ER transactivation.

To examine the impact of A3B-mediated DSB and A3B-bound R-loops on gene expression, we employed the predictive algorithm rGREAT to generate two groups of genes associated with previously identified regions (Fig. 7a). The first group comprised 86 genes associated with A3B-dependent DSBs, while the second group comprised 333 genes associated with E2-induced, R-loop-proximal A3B binding sites. Applying leading-edge analysis to the expression of these two gene groups identified a significant overlap with genes up-regulated by E2 (Fig. 7b). Importantly, genes from both groups were also enriched among genes whose E2 responsiveness was dampened upon expression of the bicistronic A3B**-hUGI construct, in which the hUGI peptide primarily exerts its effect by blocking BER processing of A3B-induced lesions. (Fig. 7b). These data are consistent with a mechanism of A3B-dependent DSB formation facilitating ER-regulated gene transactivation. To further validate this hypothesis, we examined the effects of hUGI induction and A3B depletion on DSB formation, in parallel with expression of associated genes. Two groups of genes were analysed, E2-responder genes modified by A3B**-hUGI expression and those unaffected (Fig. 7c). Consistent with our model, we found that both depletion of A3B and blocking BER repair of its deamination sites prevented A3B-dependent DSBs from forming, leading to modified E2 responses in T47D (Fig. 7d) and MCF7 (Supplementary Fig. 21) cells. In contrast, no changes in E2 responses were seen for gene-associated A3B-independent DSBs despite A3B depletion or BER blockade (Fig. 7d and Supplementary Fig. 21). Collectively, our findings indicate that the processing of A3B-mediated C > U editing sites at R-loops, leading to DSB formation, is required for the E2 response of a subset of ER-regulated genes in cells.

To sum up, the results presented in this study support a model of how ER transactivation leads to DSB formation and its functional implications for gene transcription with the following steps (Fig. 8): (i) Upon ER activation, A3B is recruited to transcriptionally active enhancers and promoters via its interaction with ER. (ii) At these sites, A3B catalyses cytidine deamination on the exposed ssDNA, generating uridine lesions and initiating strand nicks. (iii) In parallel, the TC-NER endonucleases XPF and XPG, which can incise either the displaced DNA strand or the DNA:RNA hybrid, act cooperatively to introduce SSB. The convergence of these A3B- and TC-NER-mediated incisions results in DSB formation at E2-responsive enhancers. (iv) Repair of these DSBs through DDR pathways triggers chromatin remodelling and establishes epigenetic marks favourable for sustained transcription of ER target genes.

## Discussion

The direct capture and mapping of A3B sites in cancer cells have previously been challenging due to the masking effect of the proficient BER system and the TP53-dependent synthetic lethal interaction between BER deficiency and A3B enzymatic activity[29,35]. In this study, we overcame this hurdle by employing an engineered BER-deficient cell line model with A3B overexpression in the context of TP53 functional loss. Through the sequencing of multiple clones, our approach efficiently captured A3B-driven mutations, as evidenced by strand-coordinated focal TC > TT hypermutations and NMF signatures confirming known A3B characteristics. The strong correlation between these sites and regulatory elements (transcription start sites and enhancers), along with their association with transcriptional activity and early DNA replication timing, reinforces the role of A3B as a co-activator for transcription factors, particularly ER, in our investigation. Therefore, we recommend further studies using these types of models beyond ER-driven systems to explore the potential effects of A3B on additional transcription factors, particularly other nuclear receptors.

By leveraging WGS data from our BER-deficient cell model, we provided direct evidence that R-loops serve as substrates for A3B—an insight that could not have been identified through datasets from the Cancer Genome Atlas (TCGA) project and the International Cancer Genome Consortium (ICGC)[50,51]. Although open chromatin configurations alone can facilitate A3B binding in theory, our results have demonstrated distinct sequence traits, a selectivity towards transcriptional regulatory regions and a dependence on R-loop forming for A3B genomic base editing, indicating a highly regulated mode of action. However, our results also showed that not all A3B editing or binding requires R-loop formation, regardless of ER transactivation. Whether these sites point to new types of A3B substrates, R-loop-independent functions of A3B, or other regulatory mechanisms for A3B editing will serve as interesting topics for future studies.

A comprehensive understanding of A3B's genomic editing function had remained incomplete due to the above-mentioned difficulty in identifying A3B's unmodified editing sites. Previously, emphasis has been largely placed on the consequences of coding-region mutations. Here, in a systematic study of A3B's DNA editing sites, we identify R-loop-associated regulatory regions, rather than protein-coding regions, as primary substrates for A3B in breast cancer cells. Since protein-coding mutations remain important to elucidating A3B's function, our results suggest R-loop formation at coding regions may predict their accessibility to A3B's DNA editing activity. In addition, our use of SPI-seq enabled us to comprehensively map both the ssDNA and dsDNA sites occupied by A3B, providing a complete understanding of A3B's genomic binding specificity and reinforcing its primary role as a transcriptional regulator.

Our study also provides additional insights into the role of A3B in transcriptional regulation and genome stability, particularly in the context of ER-driven gene expression. While a previous study has established that A3B interacts with R-loops and contributes to transcription-associated mutagenesis, the precise mechanism through which A3B engages with R-loops and its functional consequences remain unclear[27]. Here we propose a revised model (Fig. 8) that provides direct evidence that A3B preferentially binds and edits displaced

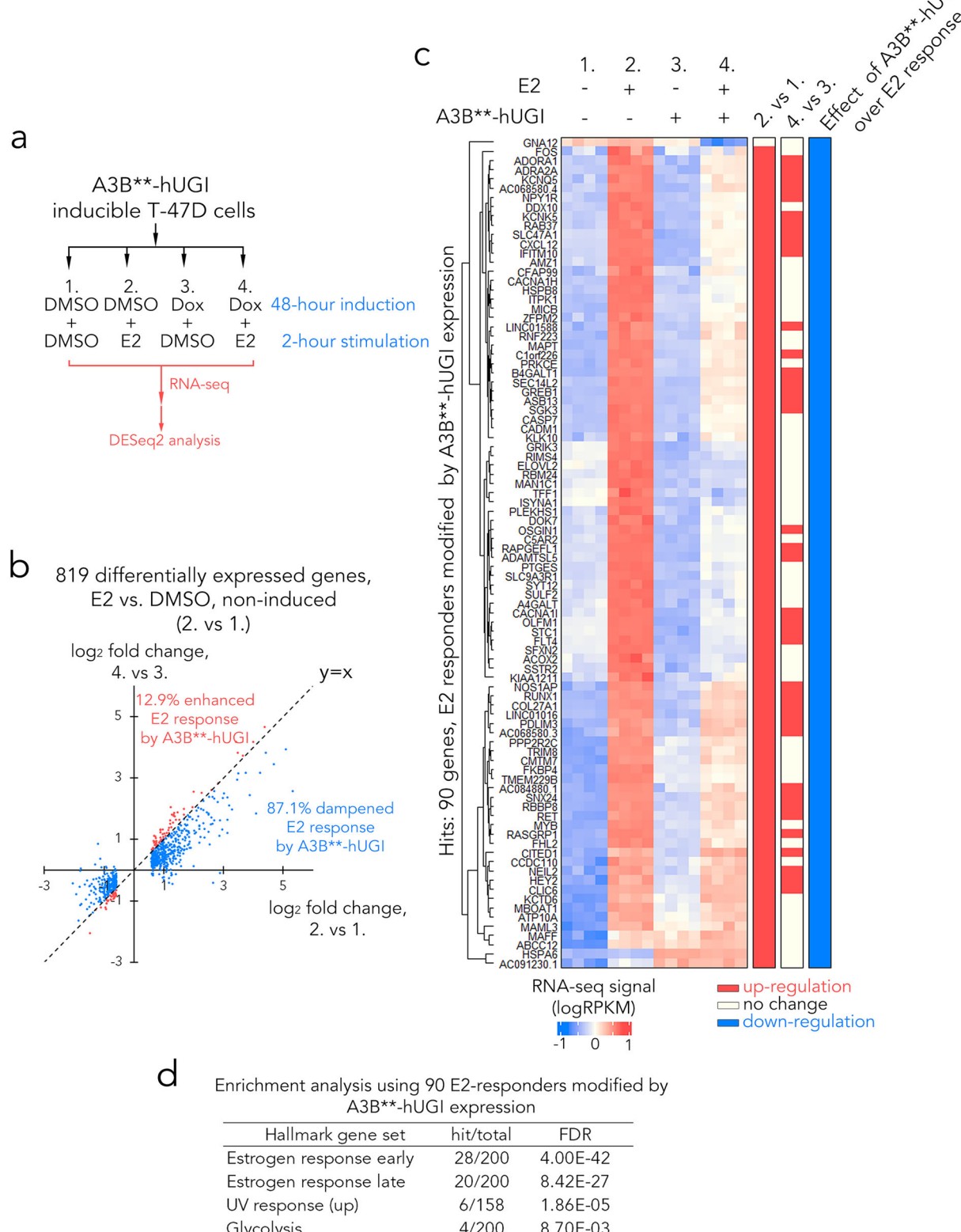

**Fig. 6 | Blocking base excision repair impairs E2 response. a** Schematic of the RNA-seq experiment in this study. For each of the treatment groups, four biological repeats were conducted for T47D cells. **b** Dot plot showing log₂ fold change of normalised transcript counts for 819 differentially expressed genes affected by E2 stimulation without the expression of A3B**-hUGI in T47D cells. **c** Heat maps showing normalised RNA-seq signals for 90 differentially expressed genes, the E2 response of which were modulated by A3B**-hUGI induction in T47D cells. Statistical results from DESeq2 are summarised in bars on the right. **d** Table showing MSigDB hallmark gene set enrichment analysis using 90 hit genes described in (**c**). Source data are provided as a Source Data file.

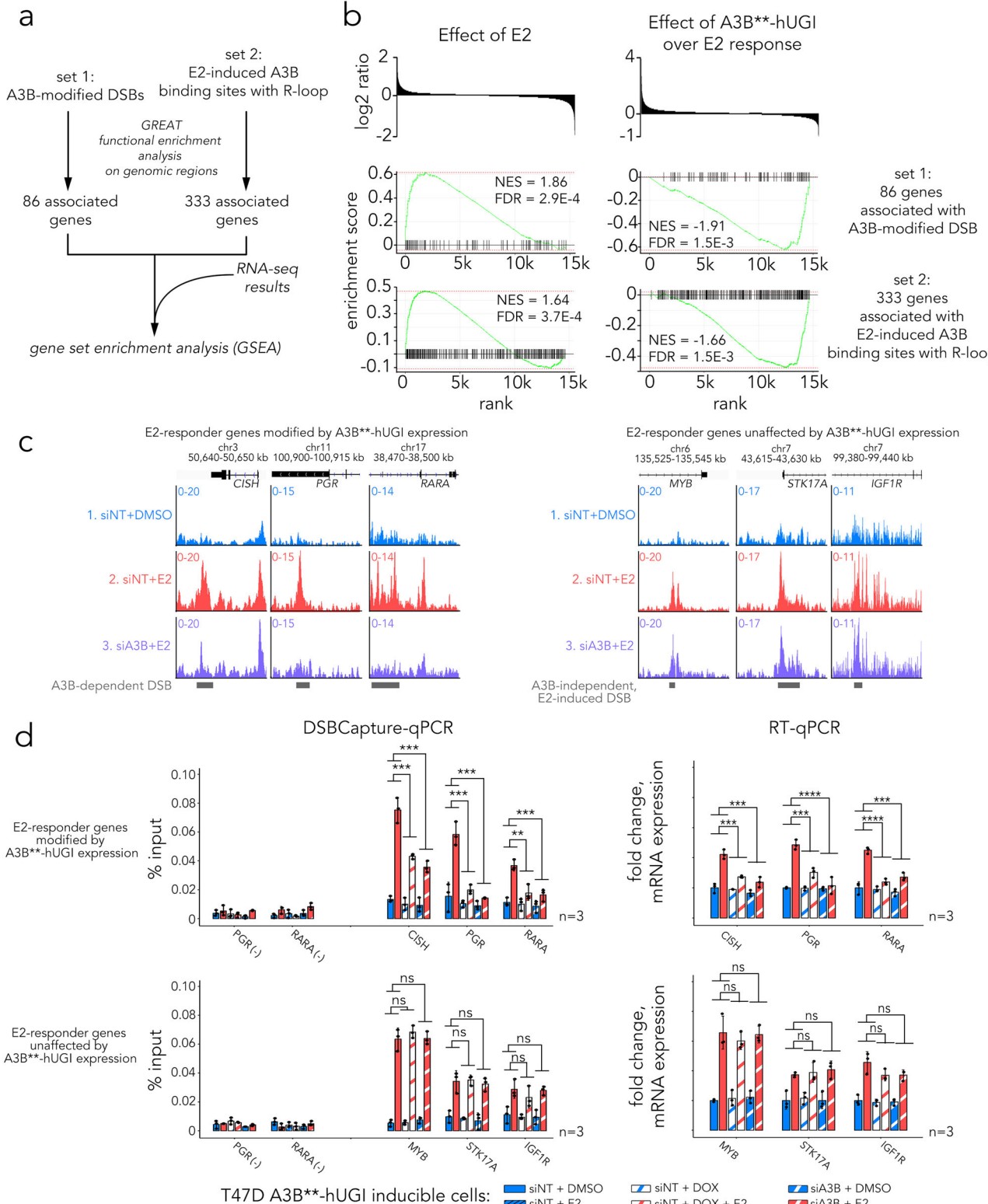

**T47D A3B\*\*-hUGI inducible cells:** siNT + DMSO | siNT + E2 | siNT + DOX | siNT + DOX + E2 | siA3B + DMSO | siA3B + E2

ssDNA within R-loops at ER promoter or enhancers. This direct coupling between A3B deamination and gene regulatory function suggests a broader role for A3B beyond its well-characterised mutagenic potential. Furthermore, our findings establish BER and TC-NER as key pathways in processing A3B-induced lesions, providing a mechanistic basis for the conversion of R-loops into DSBs

Our findings establish that R-loop formation is essential for A3B recruitment and subsequent chromatin remodelling mediated by DNA repair pathways, revealing a previously unrecognised layer of ER-

driven transcriptional control and positioning A3B as a key co-regulator of ER target genes. We show that repair of A3B-catalysed uracil lesions within R-loops leads to R-loop-mediated DSBs, which affect chromosome integrity and contribute to genomic variability in cancer cells[46,52]. Whereas previous studies proposed that removal of DNA/RNA hybrids by TC-NER endonucleases such as XPF and XPG generates strand cleavage, which is subsequently converted to DSBs by replication stress or additional nucleases [46,47,52,53]. Our data reveal an alternative, A3B-dependent mechanism. Specifically, the formation of

**Fig. 7 | Formation of A3B-modified DSBs promotes E2-regulated gene transcription. a** Diagram depicting gene set enrichment analysis (GSEA) using genes that are predicted to be associated with DSBs. Associations of genomic regions and genes were predicted using rGREAT with a binomial Bonferroni-adjusted P-value of 0.05. Leading-edge analysis were conducted using the fGSEA package with R. **b** Results for leading-edge analysis for genes associated with A3B-dependent DSB or R-loop-overlapping E2-induced Flag-A3B binding sites. **c** Signal tracks for DSBCapture at E2 responder genes either modified by A3B**-hUGI expression (*CISH*, *PGR* and *RARA*) or those unaffected (*MYB*, *STK17A* and *IGF1R*) in T47D cells. Data represents fold enrichment of signal (CPM) over the input control. **d** Bar graphs showing DSBCapture-qPCR DSB quantification (left) and RT-qPCR

transcript levels (right) in T47D cells carrying a lentiviral inducible A3B**-hUGI cassette. Cells were treated with NT or A3B siRNA, or doxycycline (DOX), for 48 h before 2 h stimulation with DMSO or 100 nM E2. Data represent the mean of three biological replicates ± SD. **: $p \leq 0.01$, ***: $p \leq 0.001$, ****: $p \leq 0.0001$, and ns: $p > 0.05$. For DSBCapture-qPCR, two-way ANOVA P-values for the effect of DOX on the E2 response are: *CISH* $2.925 \times 10^{-4}$, *PGR* $5.126 \times 10^{-4}$, *RARA* $2.763 \times 10^{-3}$; for the effect of A3B siRNA on the E2 response: *CISH* $1.25 \times 10^{-4}$, *PGR* $2.943 \times 10^{-4}$, *RARA* $6.982 \times 10^{-4}$. For mRNA expression, two-way ANOVA P-values for the effect of DOX on the E2 response are: *CISH* $2.956 \times 10^{-4}$, *PGR* $1.661 \times 10^{-4}$, *RARA* $8.584 \times 10^{-5}$; for the effect of A3B depletion on the E2 response: *CISH* $2.122 \times 10^{-4}$, *PGR* $8.351 \times 10^{-5}$, *RARA* $2.455 \times 10^{-4}$. Source data are provided as a Source Data file.

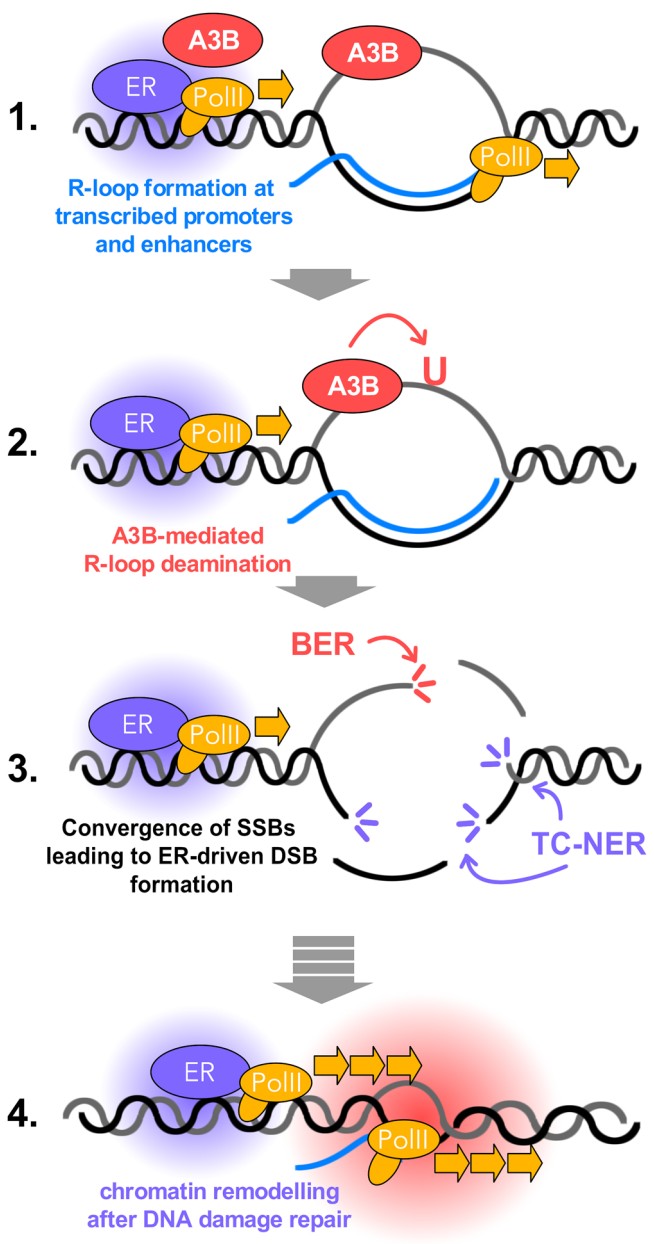

**Fig. 8 | A3B editing initiates R-loop processing following ER-activation.** Mechanistic model detailing the generation of DSBs as a result of collaborative processing of R-loops and R-loop-editing by A3B and TC-NER (A3B APOBEC3B, BER base excision repair, DSB double-strand break, ER Oestrogen receptor, PolII RNA polymerase II, SSB double-strand break, TC-NER transcription coupled nucleotide excision repair).

DSBs requires A3B for productive engagement of the TC-NER machinery, suggesting that A3B-induced deamination acts as a trigger that licences or stabilises the recruitment of NER factors to R-loop-containing chromatin. Whether A3B-modified lesions serve as direct signals for NER activation or A3B physically facilitates XPF/XPG access to structured transcriptional intermediates remains to be determined, and future work should clarify how this cooperation contributes to E2-induced genome instability.

Previous studies established that the topoisomerase TOP2B generates promoter-proximal DSBs to support E2-induced transcription and can engage XPF in processing promoter-associated R-loops[16,54]. Our data suggest that APOBEC3B acts on both promoter and enhancer R-loops, particularly at transcribed enhancers, where cytosine deamination and subsequent repair by BER and TC-NER may contribute to local chromatin remodelling associated with ER activity. While both pathways involve R-loops and related repair factors, A3B could provide a complementary, nucleotide modification-driven layer of regulation. Further work will be needed to determine whether and how these two processes intersect within E2-induced gene transactivation.

Additional investigation should also address how A3B-mediated SSBs influence R-loop homoeostasis and transcriptional outcomes. Recent findings have demonstrated that cells with high-level expression of A3B exhibit an accelerated clearance of steady-state or stimulation-induced R-loops[27]. We hypothesise that this process may also be associated with an increased incidence of DNA DSBs, as our study indicates that many A3B-induced DSBs depend on R-loop formation. Future work should also address the temporal dynamics between A3B-induced DSB formation and transcriptional activation. Because DSBs typically transiently suppress ongoing transcription, it is unlikely that A3B-mediated breaks and the associated chromatin reorganisation occur concurrently with active gene expression. We therefore propose that R-loop-induced DSBs function as upstream regulatory events in ER-driven transactivation, initiating chromatin remodelling and epigenetic modifications during or following repair to establish a chromatin environment permissive for subsequent transcriptional activation. A relevant example of this concept is the role of eRNAs, which promote gene transcription while simultaneously forming R-loops as part of their characteristic bidirectional transcription and enhancer-associated activity, which are implicated in strand breaks[55].

Recent work has also shown that paused RNA polymerase II can itself generate DNA termini that mature into double-strand breaks at specialised transcription-termination regions containing a 5'-CCTTTTTT-like pause motif downstream of mRNA polyadenylation sites[56]. Although Pol II is present at our damage sites because they are transcriptionally active, the distinct genomic localisation and strict dependence on A3B-mediated C > U editing indicate that the lesions characterised here arise through an A3B-initiated pathway rather than through a Pol II termination-associated mechanism.

Within this mechanistic context, the high prevalence of the 29 kb germline deletion eliminating the *APOBEC3B* open reading frame

(ΔA3B, rs12628403) in South-East Asian populations appears inconsistent with an obligatory physiological role for A3B in ER signalling, as homozygous carriers show no endocrine abnormalities [57,58]. Existing studies have focused primarily on cancer risk, leaving classical ER-related traits, such as pubertal timing, fertility and bone density, unexamined. Importantly, ΔA3B is not a true null: its breakpoint generates a stabilised APOBEC3A-B chimaera with elevated cytidine-deaminase activity[59], probably buffering ER pathways in normal tissues. In contrast, our data indicate that ER-positive tumours exhibit a likely oncogenic dependence on A3B, forming an amplifier circuit in which ERα induces A3B[5,60], and A3B edits R-loops at ER enhancers to accelerate transcriptional cycling. Consistent with this compensatory model, ΔA3B does not reduce breast cancer incidence and may modestly increase risk, including in ER-positive disease [57,58]. We therefore regard A3B as conditionally essential, dispensable under homeostatic endocrine signalling but strongly selected in the replication-stressed, ER-driven environment of cancer. Future work should combine rs12628403 genotyping with systematic endocrine phenotyping, clarify the respective contributions of A3A and A3B at ER enhancers, and test the therapeutic potential of targeting A3B or uracil-processing steps in genotype-stratified models.

Previous studies have also suggested A3B as a therapeutic target in the context of tamoxifen resistance, attributing resistance to increased A3B-driven gene mutations that affect protein function[24]. However, while our results indicate that A3B facilitates adaptive responses at the transcriptional or epigenetic levels in breast cancer cells, our recent characterisation of its RNA editing activity reveals that A3B-mediated RNA editing in the same model displays a stronger tendency to generate non-synonymous mutations in protein-coding transcripts[61]. Taken together, our data support therapeutic targeting of A3B in tamoxifen-resistant breast cancer because of its role in driving adaptive responses at the transcriptional or epigenetic level. Future studies dissecting A3B's dual DNA/RNA editing functions will help elucidate how its nucleic acid deamination activity alters gene expression and proteome diversity that drives cancer evolution.

## Methods

### Cell culturing and cell line preparation
T47D (ATCC HTB-133) and SKBR3 (ATCC HTB-30) cells were obtained from ATCC. Lenti-X 293 T cells were obtained from Clonetech. Cells were maintained in ATCC-recommended medium, except that Lenti-X 293 T cells were cultured in DMEM supplemented with 10% v/v FBS (PAA) and 0.5% v/v pen/strep. All cells were cultured at 37 °C with 5% $CO_2$.

To generate stable inducible cell lines, T47D cells at the exponential growth phase were transfected with lentivirus at an M.O.I of 100, with 10 μg/ml polybrene. After a 48 h transduction period, selection was performed in medium supplemented with 4 μg/mL puromycin (Gibco). Transduction efficiency was monitored by either turbo-RFP expression, A3B immuno-fluorescence microscopy or immunoblotting following induction of sample cells with doxycycline. After verification, lentivirus cassettes were maintained by culturing cells with 4 μg/mL puromycin.

### Drug treatments
For the induction of lentiviral protein expression, cells were pre-exposed to full culture medium supplemented with 1 μg/mL doxycycline (Sigma) for 24 h unless denoted otherwise. Follow-up drug treatments, if included, also contained the same doxycycline concentration throughout the entire treatment period.

For ER induction by E2, adherent cells were briefly washed with warm PBS twice before incubating with phenol-red-free RPMI-1640 medium (Gibco) supplemented with 10% charcoal-stripped FBS (Thermo Fisher) for 48 h at 37 °C. E2 was added to the culture medium at a final concentration of 100 nM, and further incubated for two hours

(unless denoted otherwise) before the cells were harvested for follow-up studies.

### Colony separation and WGS
0.25 million exponentially growing T47D cells harbouring lentiviral 3 × Flag-A3B-P2A-hUGI-HA or 3 × Flag-A3B**-P2A-hUGI-HA were seeded onto a 100 mm plate for 24 h, followed by induction with 1 μg/ml doxycycline for 120 h. Cells were harvested using trypsin digestion, serially diluted, and plated onto a 96-well plate (Corning) according to the manufacturer's protocol. Colonies derived from single cells were picked, and the cloned cells were expanded with culture medium supplemented with 4 μg/ml puromycin. Genomic DNA was extracted from each colony using a Qiagen Genomic-tips 500/G kit. Sequencing libraries for whole genome sequencing with DNA nanoball (DNB) technology were constructed (Beijing Genomics Institute (BGI) Inc., Hong Kong). Sequencing was carried out using BGISEQ-500 sequencer with a mean sequencing coverage of greater than 30 × for each of the samples using 2 × 150 bp configuration.

### Plasmids cloning and lentivirus packaging
DNA encoding 3×Flag-A3B-P2A-hUGI-HA were synthesised by GeneArt service (Thermo Fisher Scientific Inc.). In order to distinguish the mRNA of exogenous proteins from their endogenous and boost their expression, codons for both A3B and hUGI were modified using the GeneArt design algorithm. AgeI restriction site and Kozak sequences were introduced at the 5′-end, and BspD1 restriction site was introduced at the 3′-end of the construct using PCR. An empty pTRIPZ plasmid was obtained from Horizon Discovery. The DNA fragment was introduced into the pTRIPZ plasmid using AgeI and BspD1 restriction digestion and ligation with enzymes from New England Biolabs (NEB). For the A3B** construct, two rounds of site-directed mutagenesis at E68Q and E255Q sites on A3B were performed on 3 × Flag-A3B-P2A-hUGI-HA-containing pTRIPZ plasmid using the Quick-Change kit provided by Agilent. The resultant plasmids were sequenced and maintained in Mach1 T1 E. coli cells.

For lentivirus packaging, helper plasmids psPAX2 and pMD2.G were co-transfected with pTRIPZ encoding the protein of interest into Lenti-X 293 T cells (Clonetech) using a calcium phosphate transfection kit (Promega). After changing the culture medium after 24 h, the virus was harvested at 48 and 72 h, pooled, precipitated using Peg-it reagent (System Biosciences) and resuspended in serum-free RPMI medium (Gibco). Multiplicity of infection (M.O.I) of viruses was measured according to the manufacturer's recommendation (Horizon Discovery).

### Immunoblotting
Cells were lysed in RIPA buffer (Millipore) containing complete protease inhibitor cocktail (Roche) and phosphatase inhibitors solution 2 (Sigma). Protein concentrations were measured by the BCA protein assay (Pierce). Samples were separated on SDS polyacrylamide gels and transferred to PVDF membranes using the iBlot2 system (Invitrogen). Membranes were washed with TBST (25 mM Tris, 140 mM NaCl, 0.1% Tween-20, pH 7.5) and probed with various primary antibodies overnight at 4 °C, followed by binding of IRDye 680- or IRDye 800CW-conjugated secondary antibodies (LI-COR) for 1 hour at room temperature. Imaging and quantification of protein bands were performed using an LI-COR CLX infrared imaging system.

### UdgX preparation and slot blotting
DNA encoding M. smegmatis UdgX protein with N-terminal 6×His-tag and C-terminal AviTag was synthesised by GeneArt (Thermo Fisher) and cloned into the pET24a plasmid. Biotinylated UdgX protein was expressed in the AVB101 E. coli strain, and sequentially purified using Ni-NTA and Superdex75 columns on an ÄKTA Avant protein purifier.

Uracil-incorporated positive control DNA was obtained by extraction of pUC19 plasmid which was transformed into the K12 CJ236 E. coli strain (NEB). Genomic DNA from T47D cells was extracted and purified using the Qiagen Genomic-tip 500/G kit according to the manufacturer's instructions. Sample DNA (100 ng for positive control and 4 μg for genomic DNA) was loaded onto a PVDF membrane with Hoefer PR600 slot blot manifold linked to a vacuum pump. The membrane was blocked by Pierce protein-free blocking buffer (37572), followed by incubating with 1 μg/ml biotinylated UdgX at room temperature for 1 h. After probing with streptavidin-HRB conjugate (Pierce), the membranes were washed with TBST and detected with LI-COR CLX infrared imaging system using ECL-plus substrate (Pierce).

## SNV calling and analysis

Raw sequences from BGISEQ-500 sequencer were aligned to the human genome (hg38) Burrows-Wheeler Aligner (BWA)[62]. The resultant bam files were then recalibrated using GATK 4.0.5 mutation calling kit following the 'best practice' advised by the developers[63]. Briefly, bam reads were reordered, marked with group and duplication status with Picard tools, followed by base quality score recalibration by the GATK BaseRecalibrator programme. Joint variant calling was performed using GATK Mutect2 programme, with doxycycline-induced colonies as 'Tumour' sample and non-induced colonies as 'Normal' sample. In order to improve mutation calling accuracy, resources common germline variant sites from Genome Aggregation Database (gnomAD) and a 'panel-of-normal' dataset consisting of several wild-type T47D mutation calling results were included in the Mutect2 analysis[64]. After calculating the potential sampling contaminations using the GATK CalculateContamination programme, the Mutect2-derived SNVs were then selected against 14 filters implemented by the GATK FilterMutectCalls. In order to exclude SNVs that are located on either ends of homopolymer DNA sequences and within simple repeating elements, both of which are likely to generate false-positive results, the SNVs passing through GATK filters were further filtered using SNPiR software[65] with perl scripts filter_homopolymer_nucleotides.pl and pblat_candidates_ln.pl scripts only. The latter script was enabled by the pBLAT algorithm[66] and the BEDTools programme[67].

The resultant SNVs, encoded in vcf file format, were subjected to further analyses using programmes coded by R language (R Core Team). Calculation and visualisation of intermutational distance using a waterfall plot were performed with the R package 'qqman', with minor modifications to the source code[68]. Kataegis and omikli mutation clusters were detected using the hyperclust programme, as described previously[22]. The effect of variants was analysed using Variant Effect Predictor (VEP)[36], with the assistance of the genome assembly lift-over tool provided by UCSC Genome Browser[69]. Random sampling and enrichment analysis by Fisher exact test for variant effects were conducted using bedtools. For replication timing analysis for SNVs, a method described previously[22] was employed using the T47D Repli-seq dataset from ENCODE (ENCFF440QFG).

## Mutational signature analysis

All SNVs detected in the M +, A + and A- groups were used for identifying mutational signature contributions. Mutation signatures and spectrum analysis were analysed by the Bioconductor package MutationalPatterns with 30 COSMIC signatures following the standard workflow[70]. De novo mutational signature extraction was performed using the NMF algorithm[71]. Cosine similarities between de novo-extracted signatures and the COSMIC cancer mutational signatures were also calculated using MutationalPatterns. The absolute and relative contributions of each de novo signature were also determined.

## Chromatin state analysis

Chromatin state characterisation was conducted using ChromHMM software[37]. To provide modelling input, raw reads from ChIP-seq results using six chromatin marks were downloaded from the GEO database and aligned to the GRCh37 (hg19) genome assembly using BWA. The resultant BAM files were binarised and used as a training set to build the Hidden Markov Model that segments and annotates the T47D genome. To validate the chromatin model generated, the degree of overlap between known genomic features and the model was carried out using a script in ChromHMM. The genomic features were predefined by the ChromHMM authors, except that the super-enhancer regions in T47D cells were derived as previously reported[72]. For computing the overlap between ChromHMM and genomic regions of interest, the full-length peak regions were used except for SNVs, where ± 200 bp flanking regions were used.

## Strand-specific RNA/DNA immunoprecipitation (ssDRIP) and sequencing

RNA/DNA immunoprecipitation was performed on T47D cells using an optimised method previously reported by Halász et al.[73]. The entire procedure followed the best-performing experiment number 5 in the report, the detailed protocol of which was provided in the supplementary information section, except that the Covaris E220 was used for fragmenting genomic DNA with the following settings: duty factor 10%; peak incident power 75 watts; cycles per burst 1000; time 50 s; temperature 4 °C. S9.6 antibody was purchased from Kerafast (ENH002), and mouse IgG (Thermo Fisher 31903) was used as a control (if required).

ssDRIP-seq was performed using immunoprecipitated DNA:RNA hybrids. To enable strand-specificity, the product was subjected to the second-strand DNA synthesis with uracil incorporation. More specifically, the DNA:RNA hybrid was reconstituted in 1 × NEB first-strand synthesis buffer (NEB E7525) without the addition of primer and enzyme to a final volume of 20 μl. The product was immediately used as the starting material of the directional second-strand synthesis method following the manufacturer's instructions (NEB E7550). The resultant DNA was purified using SPRI beads (Thermo Fisher) at a volume ratio of 2.0 and eluted with TE buffer. After quality inspections with the BioAnalyzer 2100 (Agilent), the product was sent to BGI in Hong Kong, where strand-specific libraries were constructed using DNB technology. The resultant libraries were sequenced with the BGISEQ-500 instrument.

## Data processing of ssDRIP-seq

Pair-end raw reads were aligned to the GRCh37 genome assembly using BWA, followed by file format exchange by SAMtools. The resultant alignment reads were separated by first-in-pair strand into two files corresponding to (+) or (-) strand R-loop (or input) reads using SAMtools flag identifiers. Peak calling was performed with the MACS2 package using the pair-end fragment setting for the input and ssDRIP samples. Signal tracks were generated using MACS2, with the score representing fold enrichment relative to the input control.

For quantification of ssDRIP experiments, consensus R-loops peaks were generated firs. The resultant peaks for 12 replicates were served as input for MSPC, and recurrent peaks were filtered using parameters '-w 1E-4 -s 1E-8 -c 4'. Peaks on the same strand within 1000 bp were merged using BEDTools, and converted to GTF format using UCSC tools (bedtogenepred and genepredtogtf). Read counting was conducted using Subread, and raw counts were analysed by DEseq2 using a generalised linear model with an interaction term.

## GC skewness and G4 motif analysis

GC-skewness was calculated as previously described. Genomic GC-skew data, which is encoded as a bigwig file format with a window length of 200 bp, was created by the R-loop DB[74] and downloaded from the UCSC genome browser[69].

The DNA G4 motif was predicted using G4Hunter[75], an R package. A scan was performed using a G4 score threshold of 1.2 on the GRCh37

genome assembly, followed by composing the resulting file in bigwig format with a window length of 100 bp for follow-up studies using the R scripts provided by the authors of G4Hunter.

## Sequencing data visualisation

Bigwig files containing enrichment scores, GC-skewness score and G4 motif frequency across the GRCh37 genome assembly were used to plot the heatmaps and profile plots. The R package 'seqplots' was used to perform data visualisation via a graphical user interface[76]. For heatmaps, average data using a window length of 50 was used, whereas for profile plots, average data with standard deviation using a window length of 100 was used. Signal profiles across specific genomic regions were visualised by the Integrative Genomics Viewer (IGV)[77].

## S9.6 co-immunoprecipitation

S9.6 co-immunoprecipitation was conducted using the method as previously reported[40], except that S9.6 antibody was purchased from Kerafast (ENH002) and IgG control from Thermo Fisher (31903). Immunoblotting was carried out using S9.6-co-immunoprecipitated proteins, with HRP-conjugated secondary antibodies and SuperSignal West Femto substrate (Thermo Fisher 34094), and visualised by LI-COR CLX infrared imaging system.

## RNA/DNA hybrid slot blots

Genomic DNA from T47D cells was extracted and purified using the Qiagen Genomic-tip 500/G kit following the manufacturer's instructions. 8 µg DNA was treated with or without RNase H (NEB M0297) at a final reaction volume of 400 µl overnight at 37 °C and loaded onto a pre-wetted Hybond N+ nylon membrane (Cytiva, RPN303B) with Hoefer PR600 slot blot manifold linked to a vacuum pump. The membrane was immediately transferred to a Stratalinker UV cross-linker (model 1800 Stratagene) and crosslinked using a total energy of 120,000 microjoules. The membrane was then wetted with TBST, and blocked with StartingBlock T20 blocking buffer (37543) at 4 °C overnight, followed by probing with S9.6 antibody and anti-mouse-HRP conjugate (Cell Signalling 7076). The membrane was visualised using the LI-COR CLX infrared imaging system with SuperSignal West Femto HRP substrate (Thermo Fisher 34094).

## Protein transient over-expression and siRNA transfection

ppyCAG_RNaseH1_WT plasmid was used for transient over-expression of RNase H1 protein and was a gift from Xiang-Dong Fu (Addgene plasmid # 111906). This plasmid encodes V5-tagged human RNase H1 without the first 27 amino acids and replaced with a nuclear localisation signal. For protein transient overexpression, T47D cells were plated at a density of $5 \times 10^4$ cells per well in a 6-well plate and incubated for 24 h. After brief washing with warm PBS and replacing the medium with RPMI-1640 medium supplemented with 10% charcoal-stripped FBS, 1.5 µg of the vector was transfected into the cells by using Lipofectamine LTX PLUS (Invitrogen A12621) according to the manufacturer's instructions. Cells were then treated with DMSO or E2 and proceeded with subsequent experiments.

For RNAi experiments with siRNA transfection, the same cell culture preparation and follow-up experimental procedures were used as plasmid transfection, except that Lipofectamine RNAiMAX transfection reagent was used according to the manufacturer's instructions, and plasmid DNA was replaced with 20 nM siRNA. For DSBCapture experiments, cell culture and transfection reagents were scaled up to 100 mm plates according to the manufacturer's recommendations.

For RNAi experiments with siRNA transfection, the same cell culture preparation and follow-up experimental procedures were used as plasmid transfection, except that Lipofectamine RNAiMAX transfection reagent was used according to the manufacturer's instructions, and plasmid DNA was replaced with 20 nM siRNA. For DSBCapture experiments, cell culture and transfection reagent were scaled up to 100 mm plates according to manufacturer's recommendations.

## Chromatin immunoprecipitation (ChIP)

ChIP protocol used in this study was adapted from the protocol used by Myer's Lab at HudsonAlpha (protocol version v011014) which contributed to the ENCODE project[8] with the following modifications. Firstly, fixed T47D cells were lysed in ChromaTrap lysis buffer designed for sonication (Porvair Science, 100001), and chromatin fragmentation was carried out on a Covaris E220 instrument with optimised settings for ChIP (duty factor 2%; peak incident power 105 watts; cycles per burst 200; time 20 mins; temperature 4 °C). The resultant chromatin was digested with proteinase K (Qiagen), and the DNA concentration was measured with a Qubit fluorometer (Thermo Fisher). Secondly, Pierce ChIP-grade protein A/G beads (Pierce 26162) were used for immunoprecipitation, and chromatin binding to beads was performed at 4 °C overnight. Thirdly, for Flag-M2 antibody, optimised compositions of wash buffers are used and are as follows: 100 mM Tris pH 7.5; 500 mM LiCl; 1% NP-40; 1% sodium deoxycholate. Finally, the resultant immuno-precipitated DNA was purified by SPRI beads (Thermo Fisher) using a ratio of 2.0 and analysed using a BioAnalyzer 2100 (Agilent).

## Single-strand DNA-associated protein immunoprecipitation (SPI)

SPI was carried out following the developers' literature with modifications[41] and using the same method used to capture chromatin-bound DNA as described in the previous section, except that the resultant DNA was denatured at 95 °C and immediately quenched on ice.

## Sequencing and data processing for ChIP/SPI-seq

ChIP and SPI samples were sent to Beijing Genomics Institute (BGI Inc.) for library construction and sequencing. For analysis of ChIP samples, a standard pipeline for the construction of DNB library was used, whereas for SPI samples, the strand-specific DNB library construction protocol was requested in order to include ssDNA species in the library. Sequencing was carried out on a BGI-500 instrument with single-end 50 bp reads for both input and immunoprecipitated DNA.

For data analysis, raw reads were aligned to the GRCh37 genome assembly using BWA, followed by file format exchange by SAMtools[78]. Peak calling was carried out using the MACS2[79] package using paired input/IP sequences with modelled narrow peak settings and an extended fragment length of 200 bp. Enrichment scores were computed by MACS2 using the resultant files from peak calling and represented as a fold change of IP over control. For $n = 2$ replicated ChIP/SPI-seq data, reproducible peaks were picked using MSPC[80].

Peaks were filtering against a 'black list' region defined by the ENCODE project[8]. For quantitative analysis of the effect of E2 over DMSO on SPI-seq data, EdgeR[81] statistics were performed with the R package Diffbind[82], using a narrow peak produced by MACS2 and aligned sequence reads. To filter for the peak of interest, the top 25% peaks by base mean score from normalised sequencing read counts were included for the downstream analyses. Significant peaks were defined as FDR ≤ 0.05 with fold change ≥1.5 by Diffbind. Intersection between peak regions and Fisher's exact test with randomised genomic regions was performed by BEDTools, and data were visualised by the R package SeqPlots. Transcription factor binding motif analysis was carried out using the AME[83] programme from the MEME[84] suite. The distance between peak features was analysed by BEDTools[67], where proximity was defined as ±1.5 kb of distance. For genome-wide analysis of the ChIP-seq dataset, the R package ChIP-seeker was used.

## DSBCapture-seq

DSBCapture was conducted following a previous report[43] except that the Covaris E220 was used for fragmenting adaptor-ligated DNA with the following settings: duty factor 10%; peak incident power 75 watts; cycles per burst 1000; time 50 s; temperature 4 °C.

## Data processing for DSBCapture-seq

DNA libraries generated by the DSBCapture protocol were sequenced on an Illumina HiSeq 2500 by BGI. Raw reads were aligned to the GRCh37 genome assembly using BWA and processed by SAMtools. MACS2 was used to perform peak calling, with modelled narrow peak settings and an extended fragment length of 200 bp. Signal tracks were generated by MACS2, with the score representing fold enrichment relative to the input control. For quantification, EdgeR analysis was performed with Diffbind, using the same procedure as for ChIP-seq. The heatmap was created using the R package ComplexHeatMap[85].

## RNA-seq and quantification

Total RNA from T47D cells were extracted using the MagNA pure 96 instrument (Roche), and RNA integrity number (RIN) was determined by BioAnalyzer 2100 (Agilent). RNA libraries were constructed and sequenced by BGI using poly-dT enrichment in conjunction with DNB technology on the BGISEQ-500 instrument. Raw reads were aligned to GRCh38 genome assembly and GENCODE[86] GRCh38.p13 annotation using STAR programme[87], followed by processing of the resultant file using SAMtools. Read counting on genomic features was carried out using Rsubread[88], and the R package DESeq2[42] was used to perform statistic-based quantification. In DESeq2 analysis, in addition to terms to denote doxycycline and E2 effects in the design formula, an additional interaction term was added to denote the synergism effect between the two drugs following the user's manual written by the DESeq2 authors (http://www.bioconductor.org/packages/devel/bioc/vignettes/DESeq2/inst/doc/DESeq2.html). Expressed genes in T47D cells were defined as those with a base-mean normalised read count greater than 50% of all genes. Differentially expressed genes were picked using an FDR cut-off of 0.05 and a fold change cut-off of 1.5. The heatmap was created using ComplexHeatMap.

## Gene set analysis

Gene set analysis was carried out using MSigDB[49,89] with gene set collection H using differentially expressed genes identified by RNA-seq. Cis-regulatory region-associated genes were predicted using rGREAT[48] using differential binding peaks identified by SPI-seq and DSBCapture-seq. The resultant gene sets from rGREAT were used as input for the R package fGSE[89] to for cutting-edge analyses.

## BspH1 restriction enzyme-based biosensor assay for A3B activity

DNA cytidine deaminase activity in T47D and MCF7 cells was measured using a BspH1-based fluorescence biosensor assay[90]. Briefly, cells were lysed in ice-cold RIPA buffer supplemented with protease and phosphatase inhibitors, and lysates were clarified by centrifugation (12000 × g, 15 min, 4 °C). Protein concentration was determined by BCA assay.

For each reaction, 31.8 μL of cell lysate containing 100 μg total protein was mixed with 317.9 μL reaction buffer (50 mM potassium acetate, 20 mM Tris-acetate, 10 mM magnesium acetate, 100 ng/mL BSA, pH 7.9) in a low-binding microtube. To initiate the assay, 0.3 μL of 100 μM FAM-labelled A3B_probe substrate (5′-FAM-TA-TAAGTTATCATGATATATA-TAMRA-3′) was added and incubated at 37 °C for 1 h to allow cytidine deamination. dsDNA hybridisation and restriction digestion were achieved by adding 0.9 μL of 100 μM TAMRA-labelled complementary strand (A3B_Probe_CS) and 1 μL of BspH1 enzyme (1 μ/μL, NEB), followed by a 4 h incubation at 37 °C.

## Table 1 | Antibodies used in this study

| Antibody | Clone | involved methods | Provider | Catalogue Number |
|---|---|---|---|---|
| Flag tag | M2 | IB, co-IP, ChIP, SPI | Sigma | F1804 |
| HA tag | polyclonal | IB | Proteintech | 51064-2-AP |
| GAPDH | polyclonal | IB | Abcam | ab9485 |
| A3B | EPR18138 | IB | Abcam | ab184990 |
| A3B | polyclonal | ChIP-qPCR | Fisher Scientific | PA5-11430 |
| V5 tag | SV5-Pk1 | IB | Abcam | ab27671 |
| Top1 | monoclonal | IB | Invitrogen | MA5-32228 |
| α-tubulin | polyclonal | IB | Abcam | ab4074 |
| Vinculin | monoclonal | IB | Abclonal | A2752 |
| β-actin | monoclonal | IB | Abclonal | AC026 |
| DNA:RNA hybrid | S9.6 | ssDRIP, slot blot, co-IP | KeraFAST | ENH001 |
| CSB | E-18 | IB | Santa Cruz | sc-10459 |
| XPF | polyclonal | IB | Abcam | ab76948 |
| XPG | 8H7 | IB | Santa Cruz | sc-13563 |
| Streptavidin-HRP | n/a | IB, slot blot | Pierce | 21130 |

Fluorescence emission (500–700 nm) was recorded (excitation 492 nm) using an LS55 spectrophotometer (Perkin Elmer). Negative controls without lysate or enzyme were included for background subtraction.

## Statistics

All data were analysed using the R statistical software (R Foundation for Statistical Computing, version 4.4.1). The specific statistical test used, sample size (n), and P-values are reported in the corresponding figure legends. Unless otherwise indicated, tests were two-sided, and a significance level of $\alpha = 0.05$ was used. When extremely small exact P-values were encountered, we capped the reported value at $P < 0.0001$ and denoted this level of significance with ****.

## Datasets used in this study

Previously published datasets used in this study include: GC-skew data for GRCh19 from R-loop DB [http://r-loop.org/]; T47D ER ChIP-seq data GSE148277; T47D GRO-seq data GSE128452; T47D RNAPII ChIP-seq data GSE105793; T47D Repli-Seq data ENCFF440QFG; T47D super enhancers data GSM2862201; T47D Mnase-seq data GSE74308.

## Antibodies, siRNA oligonucleotides and constructs

The following antibodies used for this study are provided in Table 1. For siRNA, A3B knockdown was performed with the siRNA sequence reported by Periyasamy et al.[5], CSB, XPF and XPG with siRNA reported by Sollier et al.[47]. For the UdgX uridine-DNA sensor, the coding sequence corresponding to WP_011726794.1 was subcloned into the multiple-cloning site of the pET-24a expression vector. For the mammalian RNaseH I overexpression construct, the protein sequence encoded by NM_002936 was used.

## qPCR quantification

qPCR was performed on an Applied Biosystems ViiA 7 thermo cycler using POWER SYBR-Green master mix (Thermo Fisher). The primer pairs used in this study are provided in Table 2.

## Reporting summary

Further information on research design is available in the Nature Portfolio Reporting Summary linked to this article.

**Table 2 | List of qPCR primer pairs used in this study**

| Gene | Target coordinates (GRCh37) | 5'primer | 3'primer |
|---|---|---|---|
| CISH | chr3:50642686-50643064 | ACCTGGAGGAAGCGTGCCATC | AGCCACGTGCCTTCCCTGTTAC |
| PDPR | chr16:70170509-70170872 | AGAGGTGTGTGCCAAGACATCTGG | TTCAAGACCAGCCTGGCTAACATG |
| PGR | chr11:100904669-100905087 | TTCTGGGACTAGGCCAGCAGTC | AAGCTTGTCCGCAGCCTTATGC |
| RARA | chr17:38478476-38478879 | AGCACAAAAGGCAGGGGAGAAG | AGGCAAGCAAGGTCCCAACTG |
| AGR3 | chr7:16919998-16920357 | ACCATGTTGGCCAGGCTGATC | AGCACTTTGGGAGGCCGAAGC |
| CABLES1 | chr18:20840360-20840777 | CTGAACAGCTGGCCCCTTGC | AAGCTTGCAGCAGGGCAGAAAG |
| PGR- | chr11:101002083-101002454 | TCAGGACAGCATTGCCAGGTAGTC | ACCTTGTGCCTCAGTTTTCCCAAC |
| RARA- | chr17:38527751-38528130 | ATTGGTCCCCCACGCTGACATG | AGCAGCTAATGGGGGCAAAGAC |
| FKBP4 | chr12:2,903,978-2,904,157 | GGGGTAGGTGGGTCAGGAG | GGGGTAGGTGGGTCAGGAG |
| TNFSF8 | chr9:117,670,561-117,670,776 | ACTTAGGACAACAGTGAGGTGA | ACAGTGTAAGAGGGCTTTCGG |
| RMST | chr12:97,881,391-97,881,512 | ACTGGAGATAACGCTGTGACTG | AGCCAAGGGAATGGACTCATG |
| MYB | chr6:135,532,339-135,532,691 | TGGATGCCTGCACACTCATT | TCTATCTTTGCTGGCCTGCC |
| STK17A | chr7:43,622,456-43,622,703 | GAAAGCTGCACCTTCTCCCC | AGGCAGACTACCGGTAGCTC |
| IGF1R | chr15:99,395,783-99,395,999 | CACCAGAGCAGAGGAGGAGT | AGACATGGAGGGAAACGTGT |

## Data availability

The WGS, ChIP-seq, SPI-seq, ssDRIP-seq, DSBCapture-seq, and RNA-seq data generated in this study have been deposited in NCBI's Gene Expression Omnibus (GEO) database under the GEO series accession code GSE193234. The processed RNA-seq data generated in this study are provided in Supplementary Data 1 Source data are provided in this paper.

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

## Acknowledgements

This work is financially supported by Cancer Research UK (C309/A11566 and C2739/A22897) and ICR (London, United Kingdom). C.Z. was sponsored by the Science and Technology Commission of Shanghai Municipality (23S11901100). P.W. and C.Z. acknowledge additional grant support from the Wellcome Trust (212969/Z/18/Z and 094885/Z/10/Z), Cancer Research UK (C35696/A23187). P.W. is a CRUK Life Fellow and acknowledges support from CRUK Strategic Award C35696/A23187 and Infrastructure Award C309/A27413, and funding for the CRUK Children's Brain Tumour Centre of Excellence (C9685/A26398/RG93685); Wellcome Trust (Biomedical Resource and Technology Development Grant 212969/Z/18/Z to support the Chemical Probes Portal), Chordoma Foundation, Marcus Foundation, Mark Foundation, Bone Cancer Research Trust, CRIS Cancer, and The Institute of Cancer Research. Z.B. was supported by the Science, Technology, and Innovation Commission of Shenzhen Municipality (ZDSYS20200811142605017). We thank Prof. Ping Yuan from Sun Yat-sen University, Dr. Mike Walton and Dr. Alexandra Vasile from ICR for helpful discussions.

## Author contributions

C.Z., O.W.R., P.A.C. and P.W. conceived and designed this study. C.Z., Y.L., Q.Z. and M.T. conducted the experiments and analysis unless otherwise noted. B.C., Z.B., K.M. and B.A.-L. conducted analysis of sequencing data and provided guidance on data analysis using sequencing data. C.Z., A.H., M.P.L., O.W.R., P.W. and P.A.C. prepared the manuscript.

## Competing interests

C.Z., A.H., M.P.L., O.W.R., P.A.C., K.M., M.T., and P.W. are current or past employees of The Institute of Cancer Research, which has a commercial interest in a range of drug targets and operates a Rewards to Discoverers scheme, including A3B inhibitors, through which employees may receive financial benefits following the commercial licensing of a project. P.W. is an independent director at Storm Therapeutics, is a consultant/advisory board member at Astex Pharmaceuticals, CV6 Therapeutics, Black Diamond Therapeutics, Vividion Therapeutics and Nextechinvest; reports receiving a commercial research grant from Sixth Element Capital, Astex Pharmaceuticals, and Merck; has ownership interest in Storm Therapeutics, Chroma Therapeutics, and Nextechinvest; and has an unpaid consultant/advisory board relationship with the Chemical Probes Portal. The remaining authors declare no competing interests.
