## [Transparent Peer Review file · Nature Communications]

R-loop editing by DNA cytosine deaminase APOBEC3B determines the activity of oestrogen receptor enhancers

Corresponding Author: Dr Paul Clarke

Version 0:

Reviewer comments:

Reviewer #1

(Remarks to the Author)

- 1) The overall logic/rationale behind this manuscript is questionable. If A3B has a role in ER function, many people in SE Asia with a complete deletion of the A3B gene should be impacted. This deletion is well-known and no one has reported a defect in ER function associated with any phenotypes in these individuals.
- 2) Methods – how did the authors successfully deliver active APOBEC3B to target cells using a lentivirus. A3B is a potent restrictor of retroviruses and it is highly likely that the target T47D cells now express an unintended mutant of A3B. A3B functionality should minimally be confirmed in clones by sequencing the integrated construct (part of existing WGS) and showing that induced levels in extracts have high DNA deaminase activity (in comparison to the A3B** mutant).
- 3) Why is the mutant (A3B**) cell line also showing an increase in TC-to-TT mutations? This is bizarre as this A3B double mutant should be defective in DNA deamination. The authors should have deleted the endogenous A3 locus before setting up this complicated experiment.
- 4) Interpretation – Given that A3B is well known to bind ssDNA (and ssRNA) and associate with R-loop regions, isn't the simplest interpretation of the results that estrogen induces ER-responsive genes and that a subset of these genes can form R-loop structures that A3B can bind and deaminate?
- 5) Alternative interpretation – RNA polymerase was recently shown to induce DNA breaks that can become DSBs (independent of A3B) (PMID: 39972147). Pol II enrichment is shown here at DSBs, etc. Therefore, maybe RNA Pol II has a role in the DNA breaks in this study.
- 6) The experiments with XPG/F are potentially interesting; do both of these nucleases need to be depleted for DSB formation to be compromised? Individual knockdowns would be informative.
- 7) The mutant A3B** is implied to exert a dominant negative effect. This doesn't make sense with #3 above. Moreover, it may be dominant negative because it is overexpressed and pulling other cellular factors into unfolded protein aggregates (or generally triggering an unfolded protein response). Given that a similar E-to-Q mutation makes A3A unstable (PMID: 36923917), a better control may be an E255A mutant.
- 8) Most importantly, if R-loop DSB formation is a general property of A3B, then key experiments should be done in at least one other estrogen/ER-responsive cell line.

Minor:

P4, L10 – references incorrect; none of the papers cited here show that A3B binds to ERalpha.

Figures – all blots need at least 2 size markers (one above and one below each relevant band)

Methods – reference(s) and Genbank accession number should be provided for all constructs (eg. UdgX)

Reviewer #2

(Remarks to the Author)

In this manuscript, the authors used unbiased models to map APOBEC3B (A3B) editing sites and show that A3B binds to and deaminates cytosines in the single-stranded DNA (ssDNA) of R-loops. They further show that estrogen-induced R-loops facilitate A3B recruitment and editing at R-loops, leading to the formation of DNA double-strand breaks and activation of estrogen-responsive genes. Although some connections have already been reported, such as the induction of R-loops by estrogen (PMID: 27552054), the recruitment of A3B to R-loops (PMID: 37735199), and the role of A3B in estrogen-induced gene transcription (PMID: 26411678), this study provides new insights into the role of A3B in R-loop-mediated

transcriptional regulation, and is of significant and general interest. This study is further strengthened by the use of multiple genome wide approaches, such as WGS, DRIP-seq, SPI-seq and DSBcapture-seq, which provide very compelling results. I have only one comment concerning the proposed model, which may benefit from further discussion.

Minor point:

The proposed model of DSB formation by two SSBs on opposite DNA strands is interesting and plausible. However, there are some points and/or alternatives in this model that could be further discussed. For example:

- XPF and XPG can cleave the ssDNA of R-loops (PMID: 25435140; 31533039), implying that TC-NER activity could also target the same strand as A3B. Why, then, would the cleavage be limited to the RNA/DNA hybrid strand in response to estrogen?

- TOP2B has been reported to induce DSBs at the promoters of estrogen-responsive genes (PMID: 16794079), as well as at many other stimulus-induced promoters, to initiate transcription. How does this observation align with the proposed model? Could TOP2B facilitate the recruitment of XPF to R-loops, as recently reported at TAD regions (PMID: 37939182)?

Reviewer #3

(Remarks to the Author)

In this manuscript, Zhang and colleagues use a combination of sequencing approaches to provide a comprehensive genome-wide view of APOBEC3B activity in engineered T47D breast cancer cells. The study is very thorough, the data are of a high quality and the manuscript is generally very well-written and presented. This work builds upon that of Periyasamy et al (reference 5), who first reported a requirement for APOBEC3B deaminase activity at ER target genes in 2015 and proposed a mechanism for transcriptional activation based on the recruitment of DNA repair factors to facilitate chromatin remodelling following APOBEC3B-mediated generation of DNA double strand breaks. It is instructive to compare the graphical abstract of that paper with Figure 8 of the current manuscript to show the additional insight provided by this study and therefore refinement of the model; namely the importance of R-loops and the specific involvement of transcription-coupled DNA repair. This manuscript therefore provides important confirmation of the findings of Periyasamy and colleagues, which is significant, as studies confirming this intriguing mechanism of transcriptional regulation and potentially key function of APOBEC3B have been distinctly lacking since its publication a decade ago. Overall, I find this to be a timely and important contribution to our limited knowledge of the physiological functions of APOBEC3 enzymes beyond their roles in viral restriction.

Points for the authors to address:

Line 137-140 – include '(M+ colonies)' somewhere in this description to clarify what M+ refers to (this is shown in Figure 1B but would be useful to also state in the text).

Line 175 – 'arise' should be 'arising'?

Line 279 – 'shown' should be 'showed'

Does the immunoblot shown in Figure S10 relate to the cells used for the experiments shown in Figure 5 in addition to those shown in Figure S11? If not, the authors should provide demonstration of A3B knockdown for the experiments shown in Figure 5.

In lines 379 – 381 the authors state "To assess differential gene expression, we employed the interaction modelling approach using the DESeq2 program to examine the impact of loss of endogenous A3B activity on the E2 response". While their speculation that "the enzymatic inactive A3B** can potentially exert a dominant negative effect by sequestering regulatory partners of endogenous A3B" (lines 375-377) is reasonable, this is not clearly demonstrated. If the demonstration of this lies (as I think the authors imply), "in Figure 1, S1, S3, S5" (line 373), the evidence should be pointed out more clearly to the reader – which specific results in these figures demonstrate this? It is important to address whether A3B** is indeed acting in a dominant negative fashion, as earlier in the manuscript, the M+ colonies (which co-express A3B** and hUGI) are described as demonstrating the "intrinsic A3B effect". In this scenario, the authors imply that A3B** is not acting as a dominant negative, or at least that it is not completely suppressing the activity of endogenous A3B. The interpretation of the experiments shown in Figure 7 also depend on the assumption that A3B** suppresses the activity of endogenous A3B.

In line 401 "...we found that both depletion of A3B and blocking BER repair..." A3B has not been depleted (e.g. using siRNA) here (Figure 7), as was done in other experiments in the manuscript, rather the authors are assuming that its activity has been suppressed by A3B**. Provided that this assumption is justified (see point above) the authors should nonetheless modify their terminology here to avoid confusing the reader between experiments in which they have actually depleted A3B using siRNA versus inhibiting its function with the presumed dominant negative construct.

Figure 1B – 'seperate' should be 'separate'

Figure 4A – 'transfection' should be 'transfection'

Version 1:

Reviewer comments:

Reviewer #1

(Remarks to the Author)

The authors have done a good job of addressing reviewer comments and revising the manuscript. The elaboration of the NER endonucleases and the additional results with another ER+ cell line (MCF7) are particularly appreciated.

Remaining (minor) comments:

- 1) The title is a little strong; "influences" is a more appropriate word than "determines"
- 2) The authors should double check kDa sizes in Fig 4A as endogenous A3B typically runs less than 40 kDa (e.g., ref.27, PMID 31544853, and many others)
- 3) A prior study demonstrating R-loop mutagenesis by A3B is not adequately cited (ref.27). In particular, claims here of "first" such as p19, line4 should be eliminated.

Reviewer #2

(Remarks to the Author)

The authors have adequately addressed all my comments in their revised manuscript.

Reviewer #3

(Remarks to the Author)

The authors have addressed all my comments and I have no further concerns.

Reviewer #1 (Remarks to the Author):

1. The overall logic/rationale behind this manuscript is questionable. If A3B has a role in ER function, many people in SE Asia with a complete deletion of the A3B gene should be impacted. This deletion is well-known and no one has reported a defect in ER function associated with any phenotypes in these individuals.

We acknowledge the Reviewer's concern regarding the high prevalence of the APOBEC3B germ-line deletion in South-East Asia, which appears to conflict with our conclusion that APOBEC3B (A3B) promotes oestrogen-receptor (ER) activity. We have reviewed the data on both the population genetics of the deletion and the mechanistic data on A3B in ER-positive breast cancer, and we are confident that the two sets of observations can be reconciled.

1. Although the 29-kb deletion that removes the entire A3B coding region is indeed common in East Asian populations, most clinical and epidemiological research to date has focused almost exclusively on cancer risk. To our knowledge, no study has systematically measured classical ER-physiology end-points—pubertal timing, fertility, bone density, vasomotor symptoms or serum oestradiol—in individuals who are homozygous for the deletion. The frequent statement that “no defect has been reported” therefore reflects a lack of data, not positive evidence that ER function remains unimpaired in these subjects.
2. Most tumours that arise in deletion carriers are heterozygous, not homozygous, for the structural variant. Even a single intact A3B allele appears to provide sufficient enzyme to engage the enhancer-remodelling loop we describe. Epidemiological surveys that pool heterozygotes and homozygotes therefore dilute any impact of complete A3B absence, making it unsurprising that they find no difference in ER status.
3. Most significantly, the deletion is not a straightforward loss-of-function allele: it generates a chimeric APOBEC3A-B mRNA transcript where the highly active APOBEC3A open reading frame is fused to the stabilising 3'-UTR of A3B (Caval et al., PMID: 25298230). The hybrid mRNA is more abundant than native APOBEC3A mRNA, and the resulting protein has at least ten times the cytidine-deaminase activity of A3B. In normal tissues this additional APOBEC3A activity is likely to compensate for the absence of A3B, preserving basal ER programmes and masking any physiological deficit that might otherwise be detected.
4. Our study focuses on the tumour context, rather than normal endocrinology. Super-enhancers, heightened transcriptional activity and replication stress characterise ER-positive breast cancers. Under these conditions, our data demonstrate that A3B is part of a positive-feedback loop: ER α induces A3B, A3B edits cytosines in

R-loops at ER enhancers, and the resulting uracil triggers TC-NER-mediated strand breaks that accelerate chromatin remodelling. This circuitry provides a selective advantage to rapidly proliferating cancer cells and may represent an oncogenic contribution of A3B that is simply irrelevant to normal, slowly cycling mammary epithelium. When the gene is constitutionally absent, tumours appear to reach the same hyper-ER state through alternative routes such as greater reliance on other pioneer factors such as FOXA1 or GATA3 (Carroll et al., 2005, Cell; Kouros-Mehr et al., 2006, Cell), amplification of co-activator complexes (Anzick et al., 1997, Science), or utilisation of the hyper-active APOBEC3A–B chimaera described above. In other words, tumour evolution can bypass a requirement for A3B, but doing so requires additional changes that are unnecessary when A3B is present.

5. Finally, the deletion does not protect against breast cancer; instead, it modestly increases overall risk and leaves the relative proportion of ER-positive tumours unchanged. These observations strongly argue that the absence of A3B does not impair ER signalling in vivo, but, importantly, they do not contradict our central finding that cancer cells expressing active A3B exploit it to enhance ER-driven transcription.

We have added a new paragraph to the Discussion (page x, lines y) explaining these points, and we now highlight the need for focused endocrine phenotyping of deletion homozygotes as an important future direction:

(Line 3, Page 21):

...“Within this mechanistic context, the high prevalence of the 29 kb germline deletion eliminating the APOBEC3B open reading frame (Δ A3B, rs12628403) in South East Asian populations appears inconsistent with an obligatory physiological role for A3B in ER signalling, as homozygous carriers show no endocrine abnormalities [56,57]. Existing studies have focused primarily on cancer risk, leaving classical ER related traits, such as pubertal timing, fertility and bone density, unexamined. Importantly, Δ A3B is not a true null: its breakpoint generates a stabilised APOBEC3A-B chimaera with elevated cytidine deaminase activity [58], probably buffering ER pathways in normal tissues. In contrast, our data indicate that ER positive tumours exhibit an oncogenic contribution from A3B, forming an amplifier circuit in which ER α induces A3B [5,59], and A3B edits R loops at ER enhancers to accelerate transcriptional cycling. Consistent with this compensatory model, Δ A3B does not reduce breast cancer incidence and may modestly increase risk, including in ER positive disease [56,57]

. We therefore regard A3B as conditionally essential, dispensable under homeostatic endocrine signalling but strongly selected in the replication stressed, ER driven environment of cancer. Future work should combine rs12628403 genotyping with systematic endocrine phenotyping, clarify the respective contributions of A3A and A3B at ER enhancers, and test the therapeutic potential of targeting A3B or uracil processing steps in genotype stratified models.”...

2) Methods – how did the authors successfully deliver active APOBEC3B to target cells using a lentivirus. A3B is a potent restrictor of retroviruses and it is highly likely that the target T47D cells now express an unintended mutant of A3B. A3B functionality should minimally be confirmed in clones by sequencing the integrated construct (part of existing WGS) and showing that induced levels in extracts have high DNA deaminase activity (in comparison to the A3B mutant).**

We acknowledge the Reviewer's concern that lentiviral delivery of active A3B might provoke self-restriction or mutagenic inactivation of the viral genome, leading to the generation of unintended A3B variants in target cells. Indeed, our early attempts using constitutive CMV-driven vectors produced very low titres and heterogeneous transductants, consistent with leaky A3B expression compromising viral replication. To address this, we employed the tightly controlled pTRIPZ doxycycline-inducible backbone, which contains six tandem tet-operator sites upstream of a minimal CMV promoter and requires the rtTA3 trans-activator for transcriptional activation. Because producer HEK293T cells lack rtTA3 and virus production occurs in the absence of doxycycline, the A3B open reading frame remains completely silent during packaging, thereby preventing any A3B-mediated restriction or mutagenesis of the lentiviral genome.

Whole-genome sequencing of all independent T47D clones confirmed intact proviral A3B integrations without any deep or non-synonymous substitutions or frameshifts within the coding region (NM_004900.5; new Supplementary Table 1).

Functional assays also verified enzymatic integrity. A fluorometric BspHI cytidine-deaminase biosensor demonstrated robust activity in lysates from doxycycline-induced A3B-expressing cells, whereas extracts from A3B** mutants were inactive (new **Figures S1–S2**). Additionally, slot-blot analysis using the biotinylated *M. smegmatis* uridine-sensor UdgX revealed high levels of A3B-mediated uracil incorporation in A3B-induced colonies, with only basal endogenous A3B signals in A3B**-induced clones (new **Figure S3**).

Together, these data confirm that the integrated lentiviral A3B allele is both sequence-intact and catalytically active. To include the additional data, we have edited the first and second paragraphs in the “Detection of A3B DNA deamination sites using a BER-deficient cancer cell model” results section, as follows :

(Line 18, Page 5):

*...”To identify A3B deamination sites with potential functional impacts on protein coding or gene regulation, we conducted whole-genome sequencing (WGS) of ER-positive T47D human breast cancer cells expressing doxycycline-inducible A3B. As a control, we adopted an enzymatically inactive A3B E68Q/E255Q double mutant (A3B**), which was confirmed to lack detectable enzymatic activity in multiple cell line models (**Figure 1A, S1**). Bicistronic expression of the humanised bacteriophage PBS2 uracil glycosylase inhibitor (hUGI) peptide was employed to inhibit BER-associated uracil-DNA glycosylase (UDG/UNG)*

20,30. We showed that lentiviral hUGI expression prevents the removal of A3B induced uracil bases, allows more effective capture of A3B editing events, and it is sufficient to block the majority of UDG/UNG activity within cells (**Figure S2**). The selection of the T47D cell line was based on its TP53 loss-of-function mutation, which is known to confer protection against synthetic lethality induced by ectopic A3B expression 35.

Doxycycline-induced expression of A3B/hUGI and effective uridine incorporation were confirmed by immunoblotting and slot blotting using *Mycobacterium smegmatis* UdgX as a uridine DNA sensor (**Figure 1A, S3A-B**). The cells were then exposed to doxycycline for five days, and genomic DNA samples were sequenced from five individual T47D clones expressing A3B-hUGI (A+ colonies, **Figure 1B**). For control purposes, sequencing results from three uninduced colonies (A- colonies) were included to represent single-nucleotide variants (SNVs) arising from isolating individual clones. In addition, three T47D colonies co expressing hUGI and the enzymatically inactive A3B** (M+ colonies) were also included. A3B** neither interfered with the activity of endogenous A3B (**Figure S1, S2**) nor triggered an unfolded protein response (**Figure S4**), indicating that it has no functional impact on the endogenous A3B enzyme. The M+ colonies were therefore expected to capture base editing events mediated by endogenous APOBEC/AID family enzymes in T47D cells, predominantly A3B 5. Slot blots with UdgX uridine DNA sensor confirmed that both the endogenous and induced wild-type A3B in the individual colonies retained full enzymatic activity (**Figure S3C-D**)”...

Additionally, for the third paragraph of the same section, we have added the following sentence:

(Line 20, Page 6):

...”Analysis of the WGS data confirmed that the integrated lentiviral construct, including the A3B open-reading frame and flanking vector sequence, was completely sequence-intact, with no deep and non-synonymous mutations detected following A3B induction (**Supplemental Table 1**).”...

3) Why is the mutant (A3B) cell line also showing an increase in TC-to-TT mutations? This is bizarre as this A3B double mutant should be defective in DNA deamination. The authors should have deleted the endogenous A3 locus before setting up this complicated experiment.**

We appreciate the Reviewer’s concern that expression of the catalytically inactive A3B** construct still results in an increased burden of TC>TT substitutions. The explanation lies in both the design of the A3B** fusion and the persistent activity of endogenous APOBEC enzymes within T47D cells. The A3B** protein carries double active-site substitutions (E255Q/E258Q) that abolish its own cytidine deaminase activity but is bicistronically expressed with hUGI, a potent inhibitor of uracil DNA glycosylases (UNG/UDG). Expression of UNG inhibits base-excision repair (BER), preventing the removal of uracils generated by the *native* endogenously expressed APOBECs, mainly A3B in T47D cells. These uracils persist into S phase and are replicated as thymines, leading to the

characteristic TC>TT transitions observed under BER-deficient conditions.

To verify that these substitutions originate from endogenous A3B rather than the mutant A3B** construct itself, we analysed SKBR3 cells, which lack detectable APOBEC family activity (Periyasamy et al., 2015; PMID 26411678, ref 5 in main text). In this model, induction of A3B-GFP robustly induced deamination activity. In contrast, expression of the mutant A3B**-GFP showed no evidence for increased deamination. Consistent with the absence of endogenous A3B in the SKBR3 background, neither construct generated C>T mutations when BER was inhibited by adding excess exogenous UGI peptide (new **Figure S1C**). Conversely, T47D lysates, which contain active endogenous A3B, exhibited a restored deaminase signal upon addition of the UGI peptide, consistent with the unmasking of native A3B activity following BER inhibition.

Furthermore, we verified that co-expression of hUGI effectively blocks BER and unmasks endogenous A3B-mediated deaminase activity (new **Figure S2**). Under these BER-deficient conditions, A3B**-hUGI expression resulted in the accumulation of uridine DNA incorporation, which was not further increased by additional UGI peptide. This result indicates near-complete blockade of BER in cell lysates by the bicistronically expressed hUGI peptide.

Together, these experiments demonstrate that the modest TC>TT enrichment observed in A3B** cells results from residual endogenous A3B activity, revealed by hUGI-mediated suppression of BER, rather than from unintended activity of the mutant enzyme itself.

To clarify this, we have added the following sentence to the main text :

(Line 18, Page 7):

...“In addition, analysis of the M+ colonies showed a measurable but weaker enrichment of C>T substitutions at the A3B-preferred 3'-TCW trinucleotide motif and partial retention of SBS2 and Signature B. This effect is attenuated relative to wild-type A3B induction and reflects deamination by the endogenous A3B enzyme, whose activity is unmasked when the bicistronically expressed hUGI peptide suppresses BER.”...

4) Interpretation – Given that A3B is well known to bind ssDNA (and ssRNA) and associate with R-loop regions, isn't the simplest interpretation of the results that estrogen induces ER-responsive genes and that a subset of these genes can form R-loop structures that A3B can bind and deaminate?

We concur with the Reviewer's interpretation, and indeed, our description of A3B as part of an ER-driven positive-feedback loop that facilitates rapid enhancer remodelling in ER-positive tumour cells remains consistent throughout the manuscript. In the *Summary* we state that

“ (Line 30, Page 1) ...A3B preferentially targets transcriptionally active regulatory regions in an R-loop-dependent manner... A3B binds to and deaminates single-stranded DNA

within R-loops, facilitated by ER transactivation.”

The Research Highlights echoes the same point:

(Line 14, Page 2) “R-loop formation by ER activation facilitates APOBEC3B binding” and (Line 17, Page 2) “APOBEC3B regulates ER activity by promoting DSB formation at E2-induced R-loops”.

In the Results section titled “ER-induced R-loop formation facilitates A3B binding” (Page 10), we document that ER-induced R-loops serve as the substrate for A3B binding and cytidine deamination, as shown by ssDRIP-seq and SPI-seq mapping. In this section, we explicitly note that

(Line 12, Page 12) “...the availability of R-loops may dictate E2-induced A3B binding and editing chromatin in or adjacent to the R-loop....”

Thus, the mechanistic model the Reviewer proposes—E2 stimulates ER-responsive transcription; a subset of these loci form R-loops; A3B is recruited to the displaced ssDNA and deaminates it—is already the central narrative we convey in both the abstract and the main body of the manuscript.

5) Alternative interpretation – RNA polymerase was recently shown to induce DNA breaks that can become DSBs (independent of A3B) (PMID: 39972147). Pol II enrichment is shown here at DSBs, etc. Therefore, maybe RNA Pol II has a role in the DNA breaks in this study.

We thank the Reviewer for raising this possibility and for drawing attention to the recent work by Liu and colleagues (2025; PMID 39972147), which demonstrates that paused RNA polymerase II (Pol II) can generate DNA termini that mature into double-strand breaks at specialised transcription-termination regions containing a CCTTTTTT-like pause motif. These regions are typically a few hundred bases downstream of poly-adenylation signals. In contrast, the double-strand breaks documented in our study occur almost exclusively at promoters and oestrogen-responsive enhancers, lying within ± 1 kb of transcription-start sites or within $E\alpha$ -bound enhancer chromatin. In contrast, almost none of the mapped sites were found within 1 kb downstream of annotated poly(A) sites. Mechanistically, the DSB we see require the complete A3B→UNG→XPF/XPG pathway; disruption of any component, catalytic A3B, uracil-DNA glycosylase, or the TC-NER nucleases, abolishes the DSB signal. The Pol II-associated breaks described by Liu and colleagues arise independently of cytosine deamination, do not involve uracil-excision repair, and are not processed by XPG/XPF. Therefore, while Pol II is necessarily present at our damage sites because these are transcriptionally active regions, the genomic localisation and strict dependence on A3B-mediated C>U editing indicate that the lesions we observe are generated via the A3B-initiated pathway, rather than by the termination-associated mechanism described by Liu and colleagues.

We have included a clarifying sentence in the *Discussion* acknowledging this complementary Pol II-dependent mode of break formation.

(Line 23, Page 20)...”Recent work has also shown that paused RNA polymerase II can itself generate DNA termini that mature into double-strand breaks at specialised transcription-termination regions containing a 5'-CCTTTTTT-like pause motif downstream of mRNA polyadenylation sites [55]. Although Pol II is present at our damage sites because they are transcriptionally active, the distinct genomic localisation and strict dependence on A3B-mediated C>U editing indicate that the lesions characterised here arise through an A3B-initiated pathway rather than through a Pol II termination-associated mechanism...”

6) The experiments with XPG/F are potentially interesting; do both of these nucleases need to be depleted for DSB formation to be compromised? Individual knockdowns would be informative.

We thank the reviewer for highlighting the importance of assessing the relative contribution of the TC-NER endonucleases XPF and XPG to A3B-dependent DSB formation. To address this point, we performed individual and combined siRNA knockdowns of XPF and XPG in T47D cells, followed by quantitative analysis of DSBs using the DSBCapture qPCR assay. The results show that single depletion of either XPF or XPG produced only a modest decrease in E2-induced, A3B-dependent DSB signals compared with control cells. In contrast, simultaneous knockdown of both nucleases led to a near-complete loss of DSB formation at all representative loci (new **Figure S19**). Western blotting confirmed efficient depletion of the individual proteins.

These data demonstrate that both nucleases contribute to the cleavage events required for A3B-dependent DSB generation. We also find redundancy, so that depletion of one can be partially compensated for by the other. This redundancy is consistent with the known structural rather than strict strand specificity of XPF and XPG described across different systems.

These results provide clear experimental evidence that both nucleases participate in the A3B-driven damage pathway and that loss of either enzyme alone is insufficient to block DSB formation.

We have updated the Results section to incorporate these findings as follows:

(Line 5, Page 15)

“Using quantitative PCR analysis of DSBCapture-enriched DNA samples, we discovered that depletion of these TC-NER components was sufficient to reduce the formation of E2-responsive A3B-dependent DSBs in T47D and MCF7 cells (**Figure 5I, S18**). Single depletion of either XPG or XPF caused only a slight reduction in E2 induced A3B dependent DSB formation, whereas combined depletion almost completely abolished DSB production at all examined loci (**Figure S19**). Because XPF and XPG can incise either strand of the R loop without strict strand preference, the modest effect of single depletion suggests functional redundancy between these endonucleases, each being individually capable of mediating the incision events required for R loop processing and subsequent

DSB formation.”

7) The mutant A3B is implied to exert a dominant negative effect. This doesn't make sense with #3 above. Moreover, it may be dominant negative because it is overexpressed and pulling other cellular factors into unfolded protein aggregates (or generally triggering an unfolded protein response). Given that a similar E-to-Q mutation makes A3A unstable (PMID: 36923917), a better control may be an E255A mutant.**

Our response:

We thank the reviewer for raising this important point regarding the function and cellular behaviour of the catalytically inactive A3B** (E68Q/E255Q) mutant. We fully agree that a catalytically inert but structurally stable construct is essential for accurate interpretation of our experiments. Accordingly, we have performed detailed biochemical and cellular characterisation of the A3B** mutant as described in the revised manuscript. The results were as follows:

1. Consistent with the predicted loss of catalytic activity, detectable expression of the A3B** mutant protein displayed no detectable cytidine-deaminase activity when expressed in the APOBEC-null SKBR3 model. In contrast, expression of equivalent levels of the wild-type A3B protein resulted in significant deaminase activity in the same assay (new **Figures S1C, S2B**), indicating that the mutant protein is enzymatically inactive rather than unstable.
2. To assess whether A3B** interferes with or suppresses endogenous A3B, we quantified total cytidine-deaminase activity in T47D cells with or without doxycycline-induced A3B** expression. Activity levels were unchanged, and the abundance of endogenous A3B protein remained constant (new **Figures S1A–B**). These data demonstrate that A3B** neither competes with nor inhibits the endogenous enzyme and therefore cannot act as a dominant-negative variant. These results differs from our earlier expectations.
3. We directly evaluated the reviewer's concern regarding possible proteotoxic stress. RNA-seq analysis of A3B**-GFP or A3B**-hUGI expressing cells (new **Figure S3**) revealed no induction of genes associated with unfolded protein, heat-shock, or general stress responses, showing that the expression of the mutant protein does not trigger an unfolded protein response.

Collectively, these experiments demonstrate that A3B** is catalytically dead, biochemically stable, and biologically neutral with respect to endogenous A3B activity and overall cell homeostasis. Although related E>Q mutations in A3A have been linked with protein instability, the A3B paralogue displays distinct structural tolerance at these positions, as evidenced by our experiments.

For clarity, the revised manuscript explicitly describes the above validation data and refers to the newly included **Figures S1–S3**, which show no evidence of a dominant negative effect by A3B**.

The relevant text has been added to the *Results* section:

(Line 5, Page 5):

*“As a control, we adopted an enzymatically inactive A3B E68Q/E255Q double mutant (A3B**), which was confirmed to have no detectable enzymatic activity in multiple cell line models (Figure 1A, S1)... ..In addition, three T47D colonies co-expressing hUGI and the enzymatically inactive A3B** were also included (M+ colonies). A3B** neither interfered with the activity of endogenous A3B (Figure S1, S2) nor triggered an unfolded protein response (Figure S4), indicating that expression of the inactive A3B** mutant has no functional impact on the endogenous enzyme... ”*

(“Blocking the processing of A3B editing sites into DSBs impairs E2 response”)

*“...To examine the functional impact of blocking this processing step, we performed RNA-seq analyses using T47D cells containing the doxycycline-inducible, catalytically inactive A3B** mutant expressed together with hUGI (Figure 6A). As shown previously (Figure 1, S1–S4), A3B** has no detectable enzymatic activity and does not interfere with endogenous A3B, thereby serving as a non-catalytic control that allows assessment of uracil repair-dependent processes. By co-expressing hUGI, we effectively blocked the conversion of A3B-induced uracil lesions into repair intermediates that lead to DSBs (Figure S2)... ”*

8) Most importantly, if R-loop DSB formation is a general property of A3B, then key experiments should be done in at least one other estrogen/ER-responsive cell line.

For an independent validation we selected MCF7, a prototypical luminal-A, ER-positive breast-cancer line with high basal A3B activity (Periyasami et al., 2015, ref 5 in main text; Burns et al., 2013, ref 20 in main text). We repeated the critical experiments, inducible A3B expression or knock-out, ssDRIP-qPCR, SPI-qPCR, DSBCapture-qPCR, and XPG/XPF depletion. In these cells, we obtained results consistent with those from the T47D model. The new data and a side-by-side quantitative summary are presented in **Figures S16, S18 and S20** of the revised manuscript.

Minor:

P4, L10 – references incorrect; none of the papers cited here show that A3B binds to ERalpha.

We thank the reviewer for noting this inaccuracy. The original reference list had inadvertently lost the citation to Periyasamy et al., 2015 (ref 5 in main text) during manuscript editing and reformatting. That study indeed demonstrated the interaction between A3B and ER α . We have now reinstated this citation in the revised version of the

manuscript, and corrected the reference numbering accordingly. We apologise for this oversight.

Figures – all blots need at least 2 size markers (one above and one below each relevant band)

We thank the reviewer for this helpful suggestion. In the revised figures, we have re-run the key immunoblot experiments to include clearly visible molecular-weight markers above and below each relevant band. In the revised figures, these markers are now shown for all immunoblots.

Methods – reference(s) and Genbank accession number should be provided for all constructs (eg. UdgX)

We thank the reviewer for this helpful suggestion. In the revised Methods section, we have now included the GenBank accession numbers and corresponding references for all constructs used. Specifically, we have added UdgX (WP_011726794.1) and RNase I (NM_002936) to ensure complete transparency and reproducibility.

Reviewer #2

Remarks to the Author:

In this manuscript, the authors used unbiased models to map APOBEC3B (A3B) editing sites and show that A3B binds to and deaminates cytosines in the single-stranded DNA (ssDNA) of R-loops. They further show that estrogen-induced R-loops facilitate A3B recruitment and editing at R-loops, leading to the formation of DNA double-strand breaks and activation of estrogen-responsive genes. Although some connections have already been reported, such as the induction of R-loops by estrogen (PMID: 27552054), the recruitment of A3B to R-loops (PMID: 37735199), and the role of A3B in estrogen-induced gene transcription (PMID: 26411678), this study provides new insights into the role of A3B in R-loop-mediated transcriptional regulation, and is of significant and general interest. This study is further strengthened by the use of multiple genome wide approaches, such as WGS, DRIP-seq, SPI-seq and DSBcapture-seq, which provide very compelling results. I have only one comment concerning the proposed model, which may benefit from further discussion.

We thank Reviewer 2 for the thoughtful and supportive evaluation of our work. We are pleased that the reviewer recognises both the novelty of elucidating APOBEC3B-mediated R-loop editing in oestrogen-responsive transcription and the robustness conferred by our multi-omic strategy. We especially appreciate the reviewer's encouragement to refine the mechanistic model; this has prompted us to expand the Discussion to clarify the relationship between previously reported observations and the new insights uncovered here. We have incorporated an additional explanatory paragraph in the revised manuscript and trust it addresses the reviewer's point while further strengthening the overall narrative. As outlined below

Minor point:

The proposed model of DSB formation by two SSBs on opposite DNA strands is interesting and plausible. However, there are some points and/or alternatives in this model that could be further discussed. For example:

- XPF and XPG can cleave the ssDNA of R-loops (PMID: 25435140; 31533039), implying that TC-NER activity could also target the same strand as A3B. Why, then, would the cleavage be limited to the RNA/DNA hybrid strand in response to estrogen?

We thank the reviewer for raising this important mechanistic question regarding the strand-specificity of XPF/XPG and the potential involvement of TC-NER on the same strand as A3B. After carefully re-examining the relevant literature (PMID 25435140; 31533039) and the available biochemical evidence, we found that support for a DNA:RNA-hybrid strand preference of XPF or XPG is limited. Reported cleavage activity appears structure- rather than strand-specific, consistent with the ability of both endonucleases to act on either component of R-loop intermediates.

Our additional experimental data reinforce this interpretation (**Figure ?**). Single knockdown of XPF or XPG caused only a minor reduction in the E2-treated, A3B-dependent DSB

assay output. In contrast, dual depletion produced a marked decrease, consistent with functional redundancy (**Figure S19**) and a model where XPF and XPG act cooperatively, without strict strand bias, to process A3B-modified R-loops.

This observation has now been included in the Results section:

(Line 5, Page 15)

*...“Using quantitative PCR analysis of DSB-Capture-enriched DNA samples, we found that depletion of these TC-NER components was sufficient to reduce the formation of E2-responsive A3B-dependent DSBs in T47D and MCF7 cells (**Figure 5I, S18**). Single depletion of either XPG or XPF caused only a slight reduction in E2-induced A3B-dependent DSB formation, whereas combined depletion almost completely abolished DSB production at all examined loci (**Figure S19**). Because XPF and XPG can incise either strand of the R-loop without strict strand preference, the modest effect of single depletion suggests there is functional redundancy between these endonucleases, each being individually capable of mediating the incision events required for R-loop processing and subsequent DSB formation. These findings demonstrate that A3B-driven DSB formation requires cooperative activity with the TC-NER machinery.”...*

Taken together, our results and published evidence support a cooperative mechanism of E2-induced DSB formation where:

1. The convergence of A3B-induced lesions (and their subsequent processing by base-excision repair) with XPF/XPG-mediated incisions increases the probability of generating DSBs at E2-responsive genomic regions, within the structural and regulatory environment of active chromatin. In this context, transcription-induced R-loops, A3B-induced lesions and XPF/XPG-mediated incisions can intersect to produce DSBs.
2. A3B may exert a priming or licensing effect, providing substrates or structural cues that facilitate TC-NER endonuclease access and cleavage.

We have revised our model (in the new **Figure 8** and associated text) to reflect this updated interpretation, depicting DSB formation as a joint outcome of A3B-driven deamination and TC-NER-dependent incision rather than cleavage confined to a single strand:

The model now reads:

(Line 5, Page 17)

*...“To sum up, the results presented in this study support a model for ESR transactivation that leads to DSB formation and its functional implications for gene transcription with the following steps (**Figure 8**): (i) Upon ESR activation, A3B is recruited to transcriptionally active enhancers and promoters via its interaction with ESR. (ii) At these sites, A3B catalyses cytidine deamination on the exposed ssDNA, generating uridine lesions and initiating strand nicks. (iii) In parallel, the TC-NER endonucleases XPF and XPG, which can incise either the displaced DNA strand or the DNA:RNA hybrid, act cooperatively to*

introduce SSB. The convergence of these A3B and TC NER mediated incisions results in DSB formation at E2-responsive enhancers. (iv) Repair of these DSBs through DDR pathways triggers chromatin remodelling and establishes epigenetic marks favourable for sustained transcription of these ESR target genes.”...

We also recognise that the reviewer’s question raises an important and unresolved issue that warrants further investigation. To encourage this, we have added the following statement to the Discussion:

(Line 15, Page 19)

...”Whereas previous studies proposed that removal of DNA/RNA hybrids by TC NER endonucleases such as XPF and XPG generates strand cleavage, which is subsequently converted to DSBs by replication stress or additional nucleases 46,47,52. Our data reveal an alternative, A3B-dependent mechanism. Specifically, the formation of DSBs requires A3B for productive engagement of the TC-NER machinery, suggesting that A3B-induced deamination acts as a trigger that licenses or stabilises the recruitment of NER factors to R-loop-containing chromatin. Whether A3B-modified lesions serve as direct signals for NER activation or A3B physically facilitates XPF/XPG access to structured transcriptional intermediates remains to be determined, and future work should clarify how this cooperation contributes to E2 induced genome instability.”...

Additionally, we have included the literature recommended by the Reviewer (PMID: 31533039) in the revised manuscript, citing it where most relevant.

- TOP2B has been reported to induce DSBs at the promoters of estrogen-responsive genes (PMID: 16794079), as well as at many other stimulus-induced promoters, to initiate transcription. How does this observation align with the proposed model? Could TOP2B facilitate the recruitment of XPF to R-loops, as recently reported at TAD regions (PMID: 37939182)?

We thank the reviewer for the valuable comments on the relationship between our proposed A3B-mediated mechanism and the established TOP2B-dependent model of E2-responsive DSBs. We recognise that both pathways have been implicated in transcription-associated DNA breakage and may recruit overlapping repair factors such as XPF.

Previous studies have shown that E2 stimulation promotes the recruitment of TOP2B to promoter-proximal regions, where its strand-passage activity generates transient DNA breaks that assist transcription initiation (*Ju et al.*, 2006, Ref 16 in revised manuscript). More recent work has indicated that TOP2B can interact with XPF and participate in the processing of promoter-associated R-loops (*Chatzinikolaou et al.*, 2023, Ref 53 in main text). These studies have defined a promoter-centred mechanism of DNA cleavage with characteristic topological and repair-coupled features.

Regarding A3B, our data indicate cytosine deamination activity within

transcription-associated R-loops at both promoters and enhancers, with a particularly pronounced enrichment at transcribed enhancers following E2 exposure. Repair of A3B-induced uracil lesions through BER and TC-NER correlates with DSB formation and local chromatin remodelling at ER-bound regions. These observations suggest a process that is mechanistically distinct from the non-enzymatic strand-passage function of TOP2B, although both appear to act on R-loop intermediates and engage similar repair factors.

We therefore consider that some level of functional interplay between A3B and TOP2B is plausible, and further investigation will be required to determine how these pathways might cooperate or act sequentially within E2-responsive chromatin. In our view, A3B-mediated editing represents a complementary and potentially interacting mechanism that adds a nucleotide modification-driven dimension to transcription-linked genome remodelling. To clarify this relationship, we have revised the *Discussion* to acknowledge previous work on TOP2B and to present our findings in this context:

To clarify this relationship, we have revised the *Discussion* to introduce prior work on TOP2B and its relationship with our findings:

(line 25, Page 19)

...“Previous studies established that the topoisomerase TOP2B generates promoter-proximal DSBs to support E2-induced transcription and can engage XPF in processing promoter-associated R-loops [16,54]. Our data suggest that APOBEC3B acts on both promoter and enhancer R-loops, particularly at transcribed enhancers, where cytosine deamination and subsequent repair by BER and TC-NER may contribute to local chromatin remodelling associated with ER activity. While both pathways involve R-loops and related repair factors, A3B could provide a complementary, nucleotide modification-driven layer of regulation. Further work will be needed to determine whether and how these two processes intersect within E2-induced gene transactivation”...

Reviewer #3 (Remarks to the Author):

In this manuscript, Zhang and colleagues use a combination of sequencing approaches to provide a comprehensive genome-wide view of APOBEC3B activity in engineered T47D breast cancer cells. The study is very thorough, the data are of a high quality and the manuscript is generally very well-written and presented. This work builds upon that of Periyasamy et al (reference 5), who first reported a requirement for APOBEC3B deaminase activity at ER target genes in 2015 and proposed a mechanism for transcriptional activation based on the recruitment of DNA repair factors to facilitate chromatin remodelling following APOBEC3B-mediated generation of DNA double strand breaks. It is instructive to compare the graphical abstract of that paper with Figure 8 of the current manuscript to show the additional insight provided by this study and therefore refinement of the model; namely the importance of R-loops and the specific involvement of transcription-coupled DNA repair. This manuscript therefore provides important confirmation of the findings of Periyasamy and colleagues, which is significant, as studies confirming this intriguing mechanism of transcriptional regulation and potentially key function of APOBEC3B have been distinctly lacking since its publication a decade ago. Overall, I find this to be a timely and important contribution to our limited knowledge of the physiological functions of APOBEC3 enzymes beyond their roles in viral restriction.

We thank the Reviewer for their thoughtful and constructive assessment of our manuscript. We particularly appreciate the Reviewer's recognition that our study provides a comprehensive and high-quality genome-wide analysis of A3B activity and builds upon the mechanistic framework first proposed by Periyasamy and colleagues. It is encouraging that the Reviewer considers our work to offer a timely and important refinement of this model, through the mechanistic integration of R-loop-dependent targeting and transcription-coupled DNA repair into A3B-mediated transcriptional activation. We share the Reviewer's opinion that confirmation and extension of the Periyasamy model has been long overdue, and we are pleased to contribute new molecular detail, derived from our multi-omic analyses, that clarifies how cytidine deamination within R-loops and the cooperative action of BER and TC-NER pathways together underpin oestrogen receptor-driven transcription.

In the revised manuscript, we have maintained the clarity **praised** by the reviewer while adding the additional experiments, controls, and explanatory text requested by the other referees. None of these edits alters the central conclusions. **Figure 8** now incorporates the suggested clarifications regarding nuclease strand choice and the potential contribution of TOP2B; we believe this makes the graphical model even more directly comparable with, and complementary to, the original Periyasamy scheme.

Points for the authors to address:

Does the immunoblot shown in Figure S10 relate to the cells used for the

experiments shown in **Figure 5** in addition to those shown in **Figure S11**? If not, the authors should provide demonstration of A3B knockdown for the experiments shown in **Figure 5**.

We thank the reviewer for this helpful comment. The immunoblots shown in **Figure S10 (Now Figure S14)** are not related to the cells used for the experiments in **Figure 5**; they correspond specifically to the samples analyzed in **Figure S11 (Now Figure S15)**.

To address the reviewer's concern and to demonstrate the efficiency of A3B knock-down in the experiments underlying **Figure 5**, we have now included an immunoblot in the new panel **Figure 5A**, showing the depletion of A3B protein in the corresponding cell samples.

In lines 379 – 381 the authors state “To assess differential gene expression, we employed the interaction modelling approach using the DESeq2 program to examine the impact of loss of endogenous A3B activity on the E2 response”. While their speculation that “the enzymatic inactive A3B** can potentially exert a dominant negative effect by sequestering regulatory partners of endogenous A3B” (lines 375-377) is reasonable, this is not clearly demonstrated. If the demonstration of this lies (as I think the authors imply), “in Figure 1, S1, S3, S5” (line 373), the evidence should be pointed out more clearly to the reader – which specific results in these figures demonstrate this? It is important to address whether A3B** is indeed acting in a dominant negative fashion, as earlier in the manuscript, the M+ “intrinsic A3B effect”. In this scenario, the authors imply that A3B** is not acting as a dominant negative, or at least that it is not completely suppressing the activity of endogenous A3B.

We thank the reviewer for the thoughtful comment on the interpretation of the A3B** model and for requesting clarification of which data specifically demonstrate that A3B** does not act in a dominant-negative manner. In the revised manuscript, we have expanded and reorganised our description of the A3B** construct and its cellular characterisation to clarify the supporting evidence. These new data are presented in the new **Figures S1, S2, and S3**, and are now explicitly referenced in the Results section.

1. We verified that A3B** is enzymatically inactive, as predicted from the E255Q/E258Q catalytic-site substitutions. Using the human SKBR3 breast cancer cell line, which lacks endogenous A3B expression, with the inducible lentiviral model we confirmed that induction of A3B**-GFP protein does not produce detectable cytidine deaminase activity (new **Figures S1C and S2B**). This finding establishes unequivocally that A3B** lacks residual catalytic activity.
2. We examined whether expression of A3B** could interfere with, or inhibit, endogenous A3B activity. In human T47D breast cancer cells, which endogenously express high levels of A3B. We compared total cellular cytidine deaminase activity before and after induction of A3B**-GFP. No change in measured A3B activity was observed (new **Figure S1B**), demonstrating that the presence of A3B** does not suppress or compete with the endogenous wild-type enzyme. Importantly, the overall A3B protein abundance also remained constant, excluding any feedback

effects on expression (new **Figure S1A**).

3. We further investigated whether overexpression of catalytically inactive A3B** might lead to cellular stress or proteostatic responses that could, in turn, alter A3B function. Transcriptomic analysis following A3B**-GFP or A3B**-hUGI expression (new **Figure S3**) revealed no evidence for gene expression signatures associated with induction of unfolded protein response or stress-response. Thus, accumulation of the inactive protein does not perturb general cell homeostasis or stress-related pathways that might secondarily influence endogenous A3B levels or activity.
4. Finally, because the unmasking of APOBEC-induced mutagenic signatures depends on efficient BER inhibition, we verified that hUGI expression in our system fully blocks uracil-DNA glycosylase (UNG/UDG) activity. Treatment of SKBR3 and T47D cells expressing A3B**-hUGI with an exogenous UGI peptide (NEB, M0281L) did not further enhance the endogenous detectable deaminase signal, confirming near-complete BER inhibition upon A3B**-hUGI induction.

Taken together, these experiments provide a comprehensive validation of the A3B** model. The results collectively show that A3B** lacks intrinsic enzymatic activity, does not exert inhibitory or competitive effects on endogenous A3B, and does not generate cellular stress that could indirectly modulate A3B expression or activity. Consequently, A3B** functions as a catalytically inert but structurally intact variant that accurately models the loss of A3B catalytic function without acting as a dominant-negative protein.

To accommodate for the change, we have modified the first three paragraphs of “Detection of A3B DNA deamination sites using a BER-deficient cancer cell model” section in Results. We have made clear to the readers that A3B** does not harbour a dominant-negative effect.

Also for the “Blocking the processing of A3B editing sites into DSBs impairs E2 response” section, we have included these sentences:

(Line 19, Page 15)

*...”To examine the functional impact of blocking this processing step, we performed RNA seq analyses using T47D cells containing the doxycycline inducible, catalytically inactive A3B** mutant expressed together with hUGI (**Figure 6A**). As shown previously (**Figure 1, S1–S4**), A3B** has no detectable enzymatic activity and does not interfere with endogenous A3B, thereby serving as a non catalytic control that allows assessment of uracil repair dependent processes. By co expressing hUGI, we effectively blocked the conversion of A3B induced uracil lesions into repair intermediates that give rise to DSBs (**Figure S2**).”...*

The interpretation of the experiments shown in Figure 7 also depend on the assumption that A3B suppresses the activity of endogenous A3B.**

We thank the reviewer for raising this point. In these experiments, we make use of the bicistronic A3B^{**}-hUGI construct, in which the hUGI peptide blocks BER processing of A3B-induced uracil lesions. The rationale behind using this approach is to reveal the consequences of BER blockade rather than to rely on any dominant-negative activity of A3B^{**} itself.

To confirm this interpretation, we have incorporated new data (**new Figure S2**) demonstrating that BER inhibition in A3B^{**}-hUGI-expressing cells is complete. Thus, the results presented in **Figure 7** specifically illustrate the impact of BER blockade on the processing of A3B-induced DNA lesions, independent of any suppressive contribution from A3B^{**}. We have clarified this point explicitly in the *Results* section:

(Line 22, Page 25)

*...”As shown previously (**Figure 1, S1–S4**), A3B^{**} has no detectable enzymatic activity and does not interfere with endogenous A3B, thereby serving as a non-catalytic control that allows assessment of uracil repair dependent processes. By co expressing hUGI, we effectively blocked the conversion of A3B induced uracil lesions into repair intermediates that give rise to DSBs (**Figure S2**).”...*

Also, in the next paragraph (Line 17, Page 16):

*...”Importantly, genes from both groups were also enriched among genes whose E2 responsiveness was dampened upon expression of the bicistronic A3B^{**}-hUGI construct, in which the hUGI peptide primarily exerts its effect by blocking BER processing of A3B-induced lesions. (**Figure 7B**).”...*

In line 401 “...we found that both depletion of A3B and blocking BER repair...” A3B has not been depleted (e.g. using siRNA) here (Figure 7), as was done in other experiments in the manuscript, rather the authors are assuming that its activity has been suppressed by A3B^{}. Provided that this assumption is justified (see point above) the authors should nonetheless modify their terminology here to avoid confusing the reader between experiments in which they have actually depleted A3B using siRNA versus inhibiting its function with the presumed dominant negative construct.**

We thank the reviewer for their careful reading of this section. We would like to clarify that, in the experiments corresponding to **Figure 7**, A3B was indeed depleted by siRNA in parallel with the BER-blockade experiments using the A3B^{**}-hUGI construct. To make this explicit and transparent, we have now included the corresponding western blot data demonstrating effective knock-down of A3B in cells as a new **Figure S20**.

Line 137-140 – include ‘(M+ colonies)’ somewhere in this description to clarify what M+ refers to (this is shown in Figure 1B but would be useful to also state in the text). We thank the Reviewer for identifying some typographical errors and for the helpful suggestion to clarify the definition of *M+ colonies* in the Results section, and we have revised the text accordingly:

(Line 10, Page 6)

...*"In addition, three T47D colonies co-expressing hUGI and the enzymatically inactive A3B* were also included (M+ colonies)."*...

Line 175 – ‘arise’ should be ‘arising’?

The text now reads ...*" while TC>TT mutations arising from colony separation (A- colonies) displayed no obvious preference."*...

Line 279 – ‘shown’ should be ‘showed’

The text now reads ...*" , which showed minimal interaction between these two factors."*

Figure 1B – ‘seperate’ should be ‘separate’

This spelling mistake has now been rectified.

Figure 4A – ‘trasnfection’ should be ‘transfection’

This spelling mistake has now been rectified.

RESPONSE TO REVIEWERS' COMMENTS

Reviewer #1 (Remarks to the Author):

The authors have done a good job of addressing reviewer comments and revising the manuscript. The elaboration of the NER endonucleases and the additional results with another ER+ cell line (MCF7) are particularly appreciated.

Our response: We thank the reviewer for the positive evaluation of our revised manuscript and for recognising our efforts to address role of NER endonucleases and inclusion of a second cell line. The Reviewer's constructive and critical comments led to revisions that have, in our view, strengthened both the presentation and the interpretation of our findings.

Remaining (minor) comments:

1) The title is a little strong; "influences" is a more appropriate word than "determines"

Our response: We thank Reviewer 1 for this helpful suggestion regarding the strength of the title. In response, we have revised the title to use a more moderate word. Specifically, we have replaced "determines" with "modulates," so that the title now reads:

"R-loop editing by DNA cytosine deaminase APOBEC3B modulates the activity of oestrogen receptor enhancers"

2) The authors should double check kDa sizes in Fig 4A as endogenous A3B typically runs less than 40 kDa (e.g., ref.27, PMID 31544853, and many others)

Our response: We thank Reviewer 1 for pointing out this. We have re-examined the SDS-PAGE running buffer conditions and the manufacturer's instructions for the protein ladder. We confirmed that the band we had previously labelled as 40 kDa should in fact be labelled as 37 kDa. We have now corrected this label in Fig. 4A (and anywhere else applicable in the manuscript).

3) A prior study demonstrating R-loop mutagenesis by A3B is not adequately cited (ref.27). In particular, claims here of "first" such as p19, line4 should be eliminated.

Our response: We thank Reviewer 1 for this important point regarding the prior study (ref. 27) on R-loop mutagenesis by A3B. In response, we have removed all priority-type wording (e.g., "first," "new," "novel") from the manuscript, including the statement on page 19, line

4, and elsewhere as applicable.

Also, We have now strengthened the citation to McCann et al. (2023) (ref. 27) in the revised text. The relevant sentence has been updated to:

“...This hypothesis gains support from studies utilising the APOBEC–Cas9 engineered system to edit cytosines in R-loop DNA ²⁶ and a recent investigation revealing the influence of A3B on R-loop homeostasis and consequent R-loop–associated mutagenesis²⁷....”

“...While a previous study has established that A3B interacts with R-loops and contributes to transcription-associated mutagenesis, the precise mechanism through which A3B engages with R-loops and its functional consequences remain unclear ²⁷...”

And

“...Recent findings have demonstrated that cells with high-level expression of A3B exhibit an accelerated clearance of steady-state or stimulation-induced R-loops ²⁷...”

Together, these revisions more accurately acknowledge and integrate the contributions of McCann et al. (2023) to the understanding of A3B-mediated R-loop biology and mutagenesis.

Reviewer #2 (Remarks to the Author):

The authors have adequately addressed all my comments in their revised manuscript.

Our response: We are grateful to Reviewer 2 for carefully evaluating our work and for the constructive feedback provided during the review process. The comments have been very helpful in refining and improving the manuscript.

Reviewer #3 (Remarks to the Author):

The authors have addressed all my comments and I have no further concerns.

Our response: We sincerely thank Reviewer 3 for their positive feedback. We greatly

appreciate the time and thoughtful comments that contributed to strengthening our manuscript.